# Anti-quenching NIR-II molecular fluorophores for in vivo high-contrast imaging and pH sensing

Shangfeng Wang[1], Yong Fan[1], Dandan Li[1], Caixia Sun[1], Zuhai Lei[1], Lingfei Lu[1], Ting Wang[1] & Fan Zhang [1]

The contrast and sensitivity of in vivo fluorescence imaging has been revolutionized by molecular fluorophores operating in the second near-infrared window (NIR-II; 1000-1700 nm), but an ongoing challenge is the solvatochromism-caused quenching in aqueous solution for the long-wavelength absorbing fluorophores. Herein, we develop a series of anti-quenching pentamethine cyanine fluorophores that significantly overcome the severe solvatochromism, thus affording stable absorption/emission beyond 1000 nm with up to ~44-fold enhanced brightness and superior photostability in aqueous solution. These advantages allow for deep optical penetration (8 mm) as well as high-contrast and highly-stable lymphatic imaging superior to clinical-approved indocyanine green. Additionally, these fluorophores exhibit pH-responsive fluorescence, allowing for noninvasive ratiometric fluorescence imaging and quantification of gastric pH in vivo. The results demonstrate reliable accuracy in tissue as deep as 4 mm, comparable to standard pH electrode method. This work unlocks the potential of anti-quenching pentamethine cyanines for NIR-II biological applications.

[1] Department of Chemistry, State Key Laboratory of Molecular Engineering of Polymers, Shanghai Key Laboratory of Molecular Catalysis and Innovative Materials and iChem, Fudan University, Shanghai 200433, P. R. China. Correspondence and requests for materials should be addressed to F.Z. (email: zhang_fan@fudan.edu.cn)

Fluorescent imaging and sensing with high spatio-temporal resolution and sensitivity allow the direct visualization of dynamic biological interests at different levels of components from the molecules, cells in vitro to the tissues, organs in vivo[1–6]. Disastrous light attenuation and background autofluorescence in tissue at conventional imaging window of 400–900 nm have limited this technique for in vivo analysis, but they both decrease at progressively longer wavelength[7]. Over the past decade, advances in the development of functional fluorophores operating in the second near-infrared window (NIR-II; 1000–1700 nm) have allowed the investigations of deep anatomical features in vivo with high resolution and sensitivity[8–17]. However, the shortage of high-quality NIR-II fluorophores has never been more apparent. The ideal fluorophores should have high brightness (absorption coefficient ($\varepsilon$) × quantum yield ($\Phi_{fl}$)) with well quenching tolerance for high-sensitive and fast imaging, and absorb/emit at long wavelength for deep-penetrating and high-contrast imaging. Furthermore, it is better to have flexible strategies for fluorescence modulation, so that novel functional probes can be rationally designed to report events in their environment[18,19].

Molecular fluorophores are preferable for optical bioimaging because of their low toxicity and well-defined clearance mechanisms from the body, drawing significant interests across life and medical sciences[13,20,21]. Current studies on the design of NIR-II molecular fluorophores mainly focused on two types of architectures, namely donor–acceptor–donor (DAD) fluorophores and polymethine cyanines[22]. The former show emission peaks beyond 1000 nm and impressive performance in various bioimaging applications[23–25], but require high injection dose and fluence rate limited by their weak and short-wavelength absorption ($\varepsilon \approx 10^3$–$10^4\,M^{-1}\,cm^{-1}$, $\lambda_{abs} < 900$ nm). The latter consisting of two heterocyclic terminal groups connected by polymethine linker, exhibit extremely intense absorption ($\varepsilon > 10^5\,M^{-1}\,cm^{-1}$) with tunable long-wavelength up to 1200 nm, which makes them more suitable for NIR-II imaging[26]. The advantages have been demonstrated recently that even the short-wavelength indocyanine fluorophores with only ~ 5% of the total emission tailed in the NIR-II region outperform the commercial DAD fluorophore in bioimaging in vivo[27]. Conceivably, longer wavelength cyanine fluorophores would perform better. However, such fluorophores conventionally designed by enhancing donor strength of heterocycles as well as lengthening polymethine chain (Fig. 1a) suffer from significant solvatochromism-caused quenching in polar solvents. The effect is characterized by broadening and weakening of absorption band (Fig. 1c), causing the loss of brightness and invalid excitation in aqueous solution, thus compromising their biological application. As a result, to tune wavelength of fluorophores into NIR-II range whilepossessing favorable anti-quenching capability in aqueous solution remains a great challenge to date.

Here, we introduce a series of anti-quenching cyanine fluorophores with stable peak absorption/emission up to 1015/1065 nm in aqueous solution. These cyanine fluorophores are all based on pentamethine structure, and we tune their wavelength by introducing electron-donating diethylamino moieties at different position of the terminal benzothiopyrylium heterocycles. Unlike conventional NIR-II heptamethine cyanines, these benzothiopyrylium pentamethine cyanines (BTCs) overcome the significant solvatochromism in polar solvents, and this is the key to obtain various anti-quenching properties including high brightness and superior photostability in aqueous solution. Consequently, BTCs can be effectively excited at longer wavelength with deep penetration depth up to 8 mm, and outperform their heptamethine analog IR26 and clinical-approved indocyanine green (ICG) in lymphatic imaging with higher contrast. In addition, the

substituent position of diethylamino groups show significant influence on the photophysical properties and protonation behaviors of BTCs, revealing the different delocalization effects along the π-conjugation. Interestingly, BTC1070 shows ratiometric fluorescence change from 1065 to 980 nm between pH 1–4. Thus, by dividing NIR-II ratiometric imaging sub-regions and drawing calibration curves, we realize noninvasive gastric pH evaluation at a depth of 4 mm with reliable accuracy comparable to standard pH meter. The results demonstrate the potential of BTCs to be developed as fluorescent probes for in vivo high-contrast biosensing applications.

## Results

**Molecular design and photophysical properties of NIR-II BTCs.** Several commercial NIR-II heptamethine cyanine fluorophores, including IR1048, IR1061, and IR26 (Supplementary Figure 1), show intense and sharp NIR-II absorption in dichloromethane (Fig. 1c, Supplementary Figure 2). However, their absorption spectra suffer from attenuation and broadening in polar solvent such as dimethyl sulfoxide and aqueous solution. Especially for IR26 with micelle protection, absorption coefficient at 1080 nm is almost bleached in aqueous solution. This solvatochromic behavior has been generally attributed to the symmetry breaking effect in ground/excited state, connected with the length of polymethine chain as well as the donor strength of terminal groups[28,29]. Accordingly, for the cyanine fluorophores with different terminal groups, the effective polymethine chain length for stable spectra will be generally less than seven methine units. We then speculate that shortening polymethine chain would be a direct and beneficial way to achieve more stable cyanine fluorophores for bioimaging.

As a proof of concept, we designed and synthesized a series of pentamethine benzothiopyrylium cyanines (Fig. 1d–f, Supplementary Figure 3), which exhibit feature absorption of cyanine in various polar solvents, such as dimethyl sulfoxide, methanol, and even water (Fig. 1g–i, Supplementary Figure 4). The benzothiopyrylium donors contribute to the extremely long absorption/emission wavelength, bathochromically shifting ~ 300 nm compared with classic pentamethine indocyanine fluorophores (Fig. 1j, Table 1)[30]. Introducing electron-donating diethylamino moieties on benzothiopyrylium elicits further bathochromic-shifts with significant site dependency. BTC1070 has the longest absorption/emission with maximum wavelength at 1014/1070 nm, which could be attributed to the intramolecular charge transfer (ICT) effect of nitrogen atoms[31,32]. Furthermore, owing to the shorter polymethine chain, apparent improvements on the photophysical properties of fluorophores are observed. For example, BTCs exhibit higher $\Phi_{fl}$ than their heptamethine analog IR26 (Supplementary Table 3), resulting in their higher brightness. The difference is more apparent in aqueous solution that BTCs show ~ 7- (BTC1070) to ~ 44-fold (BTC982) enhanced brightness relative to IR26, demonstrating the superior anti-quenching ability (Fig. 1k). Furthermore, BTCs also show ~ 5.7- (BTC1070) to ~ 36-fold (BTC982) brighter than ICG, which has been regarded as a bright NIR-II benchmark recently[27]. In addition, all BTCs exhibit higher photostability than IR26 and ICG (Fig. 1l), showing another anti-quenching capability for long-term continuous fluorescence imaging.

To further elucidate the influence of diethylamino substitution on the photophysical properties of BTCs, we performed TDDFT calculations at the B3LYP/6–311 G(d,p) level. As shown in Fig. 2, both nitrogen atoms of BTC982 and BTC1070 contribute to the HOMO orbitals, increasing their energy levels. The difference is that in BTC1070, the distribution of HOMO on the diethylamino group is significantly delocalized onto the p-methoxy phenyl

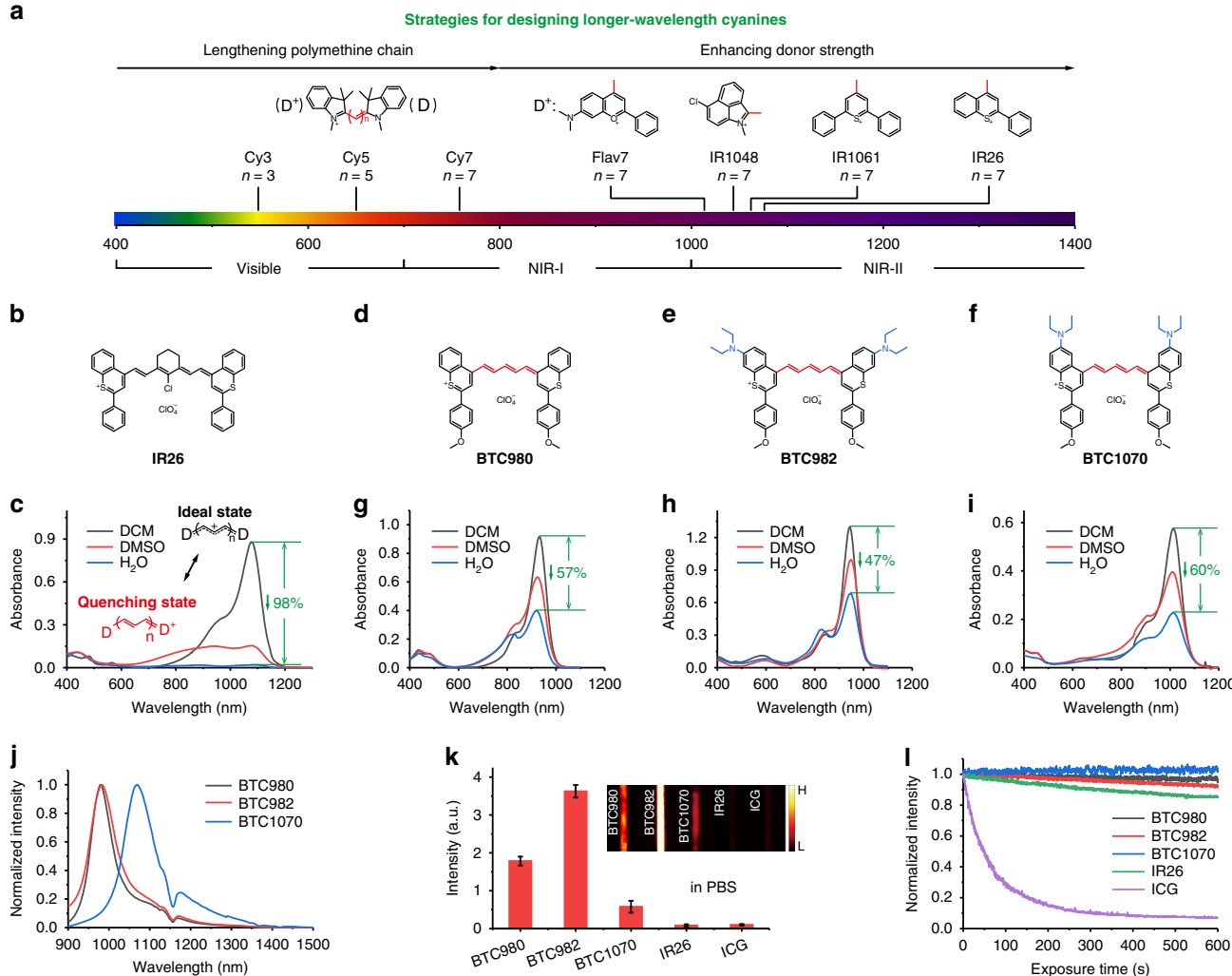

**Fig. 1** Molecular design and spectra properties of NIR-II BTCs. **a** Schematic illustration shows the general strategies for tuning absorption wavelength of cyanines: lengthening polymethine chain (n denotes the number of methine unit) leads to bathochromic shift; however, enhancing donor (D) strength by heterocycle modification is a more-effective strategy to afford NIR-II absorbing/emitting cyanines. **b**, **c** Chemical structure **b** and absorption spectra **c** of IR26 in dichloromethane (DCM), dimethyl sulfoxide (DMSO) and H₂O (phospholipid micelle formation). The ideal state, namely cyanine limit, presenting equalized charge and minimal bond length alternation in a symmetric cyanine structure, is characterized by intense and narrow absorption in nonpolar solvents such as DCM. Polar solvents such as DMSO and H₂O break the symmetry, leading to weak and broad absorption, corresponding to a quenching state called as beyond the cyanine limit. **d-f** Rational design of pentamethine benzothiopyrylium cyanines (**d**, BTC980, **e**, BTC982, and **f** BTC1070) by shortening polymethine chain (red) and introducing electron-donating diethylamino groups (blue). **g-i** Intense and sharp absorption spectra of BTC980 **g**, BTC982 **h**, and BTC1070 **i** (5 μM) in dichloromethane, dimethyl sulfoxide, and H₂O (phospholipid micelle formation). **j** Normalized fluorescence spectra of BTC980, BTC982, and BTC1070 in DCM. The ~1150 nm dips are caused by solvent absorption (Supplementary Figure 6). **k** NIR-II (1000–1700 nm) brightness comparison of equimolar (10 μM) BTC980, BTC982, BTC1070, IR26, and ICG in PBS (pH 7.4) on an InGaAs camera. Inset: fluorescence images of capillaries filled with fluorophores. Color bar ranges from 0 to 30,000 for each image. The detailed imaging parameters for each image are listed in Supplementary Table 1. Source data are provided as a Source Data file. **l** Photostability comparison of all fluorophores (5 μM) in PBS (pH 7.4) under continuous-wave laser exposure (ICG: 808 nm, BTC980: 915 nm, BTC982: 940 nm, BTC1070 and IR26: 1064 nm) at a fluence rate of 2.3 W cm⁻². The bars in k represent mean ± s.d. derived from n = 3 replicated measurements of every pixel of the capillaries

group of LUMO upon photoexcitation. This suggests that an ICT state is formed, resulting in a significant decrease of the LUMO level. Furthermore, the $S_0$–$S_1$ excitation energies from HOMOs to LUMOs verify the absorption wavelength change trend of BTCs, and the corresponding oscillator strengths correlate well with their high absorption coefficients. Surprisingly, the π-electron contribution of two nitrogen atoms in BTC982 do not obviously lower the $S_0$–$S_1$ excitation energy or bathochromically shift the wavelength. A plausible reason could be the non-coplanar geometry of BTC982 with a higher torsional angle of 29.35° compared with that of BTC980 (15.48°) and BTC1070 (18.59°) (Supplementary Figure 7).

**Optical penetration study of NIR-II BTCs in tissue phantom.** Encouraged by the above results, we then assayed the penetration properties of these BTCs in mimic biological tissue (1% Intralipid[33]) to probe the benefits of anti-quenching effect by using a home-built InGaAs setup (Fig. 3a). Taking BTC1070 with the longest wavelength as an example, capillary filled with BTC1070 micelle solution immersed into Intralipid was excited at 1064 nm and the emission was collected beyond 1200 nm at varying depths. The imaging performance was further compared with that of IR26, and benchmarked against the NIR-I (850–950 nm) and NIR-II (> 1000 nm) phantom imaging of ICG excited at 808 nm. With increase of penetration depth, attenuation of image

**Table 1 Photophysical properties of BTCs**

| Dye | $\lambda_{abs}{}^a$ (nm) | $\varepsilon^a$ (M$^{-1}$ cm$^{-1}$) | $\lambda_{fl}{}^a$ (nm) | $\Phi_{fl}{}^{a,b}$ (%) | $\varepsilon\Phi_{fl}{}^a$ (M$^{-1}$ cm$^{-1}$) |
|---|---|---|---|---|---|
| BTC980 | 932 | 184000 | 980 | 0.57/0.22[c] | 1049 |
| BTC982 | 944 | 260000 | 982 | 0.68/0.3[c] | 1768 |
| BTC1070 | 1014 | 115000 | 1070 | 0.09/0.016[c] | 104 |

[a]Photophysical properties in dichloromethane. [b]For determination of the fluorescence quantum efficiency ($\Phi_{fl}$), IR26 in dichloroethane ($\Phi_{fl}$ = 0.05%) was used as a fluorescence standard. [c]Measured in PBS 7.4 in a phospholipid micelle formation

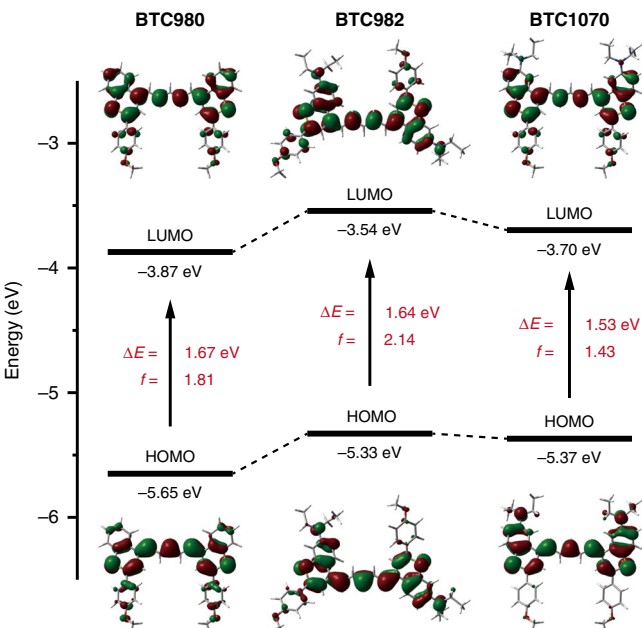

**Fig. 2** Theoretical calculations for investigation of photophysical properties. Comparison of the HOMO and LUMO energy levels, $S_0$–$S_1$ excitation energies, and oscillator strengths for BTC980, BTC982, and BTC1070, based on TDDFT calculations at the B3LYP/6–311 G(d,p) level

intensities and blurring of capillary profiles are observed for all fluorophores, but only images of BTC1070 resolve sharp edges of the capillary at a depth up to 8 mm (Fig. 3b). We define the penetration depth as the maximum imaging depth beyond which the signal-to-background ratio (SBR) of image drops below a threshold value of 2, where capillary profiles also cannot be resolved. As a result, BTC1070 has a penetration depth of 8 mm, which is fourfold deeper than IR26 (2 mm), and 8- and 2.7-fold deeper than ICG imaged in NIR-I (1 mm) and NIR-II (3 mm) range, respectively (Fig. 3c). In addition, cross-sectional profiles of capillary show clear feature integrity for BTC1070 compared with IR26 and ICG (Fig. 3d, Supplementary Figure 8), which can be attributed to the higher brightness and reduced scattering events at longer emission wavelength of BTC1070.

The longer excitation wavelength (1064 nm) of BTC1070 also improves the penetration depth, image SBR and feature contrast by decreasing signal attenuation in Intralipid. Fitting exponential function to the intensity decay curves give a much lower attenuation coefficient ($\tau$) for BTC1070 (0.6662) compared with ICG (808 nm excitation) both in NIR-I (0.9264) and NIR-II (0.9002) range, consistent with the deep penetration of 1064 nm (Supplementary Figure 9). According to the Lambert–Beer law, signal attenuation depends exponentially on the depth (optical length) of an emitting structure and the attenuation coefficient of the medium to photon. This signal attenuation coefficient is

contributed equally by the attenuation of excitation and emission photons traveled through turbid media, which is a direct result of photon scattering and absorption (Supplementary Note 1). Therefore, to eliminate the influence of different emission wavelength, phantom imaging of BTC1070 under 915 nm excitation was performed and parallelly compared with that under 1064 nm excitation at the same filter sets (Supplementary Figure 10, 11). The results also show that 1064 nm excitation gives a lower signal attenuation coefficient (0.6662 for 1064 nm excitation vs 0.7946 for 915 nm excitation) and deeper penetration depth (8 mm for 1064 nm vs 7 mm for 915 nm) (Supplementary Table 4), consistent with the light attenuation property in Intralipid (Supplementary Figure 12). In additionally, despite imaging at the same emission wavelength, the reduced signal attenuation under 1064 nm excitation gives improved image SBR and feature contrast (Supplementary Table 5). Overall, the results demonstrate the significance of developing long-wavelength absorbing fluorophores for high-contrast deep-penetration imaging.

**In vivo lymphatic imaging and photobleaching study**. To further investigate the advantages of these anti-quenching BTCs for NIR-II biomedical imaging, we compared the lymphatic imaging quality in nude mice by using BTC1070, IR26, and ICG as contrast agents (Fig. 4, Supplementary Figure 14–17). Fluorescence images were acquired 1 h after intradermal injection of contrast agents at rear paw of nude mice (Fig. 4a). Because of the severe fluorescence quenching, IR26 shows weak signal in lymphatic drainage, indicating its limitation for biological application (Fig. 4c). In contrast to the blurry lymphatic structure imaged with ICG in the short-wavelength region (850–950 nm) (Fig. 4d), much sharper images with distinguished lymph nodes as well as the afferent and efferent lymph vessels are acquired in the NIR-II range of both ICG (Fig. 4e) and BTC1070 (Fig. 4f, Supplementary Figure 17). However, crowded collateral lymph vessels are only observed by detecting longer wavelength photons of BTC1070 excited at 1064 nm (Fig. 4g). A cross-sectional intensity profile shows a minimum feature size (full-width at half-maximum) of 84 μm and maximum SBR of 9.42 (Fig. 4h). The results clearly demonstrate anti-quenching BTC1070 outperforms IR26 and ICG in lymphatic imaging.

Next, a time window with stable NIR-II signal during lymphatic drainage imaging was used to verify the superior photostability of BTC1070 in vivo via continuous laser irradiation. As a contrast, ICG is absolutely photobleached after 6 min of 808 nm laser irradiation (Fig. 4i, j). The signals of lymph nodes recover after 15 min, confirming it is indeed caused by photobleaching instead of the damage of lymphatic vessels. However, for BTC1070, after 60 min of 1064 nm laser irradiation with the same fluence rate, the lymph nodes and vessels could still be clearly distinguished (Fig. 4k), and the signal of popliteal lymph nodes is bleached only by ~ 30% (Fig. 4l). The results demonstrate the superior photostability of BTC1070, which will

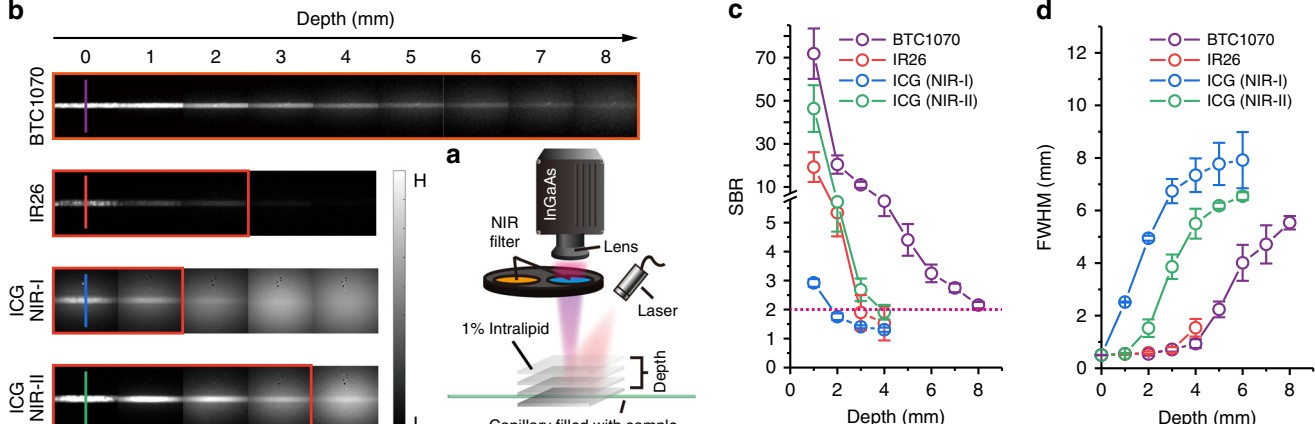

**Fig. 3** Deep penetration of bright and long-wavelength BTC1070 in mimic tissue. **a** A home-built NIR fluorescence imaging setup for tissue phantom study. **b** Fluorescence images of capillaries filled with BTC1070, IR26, and ICG in PBS (pH 7.4), respectively, immersed in 1% Intralipid with varying depth. BTC1070 and IR26 imaging signals were collected in 1200–1700 nm region under 1064 nm excitation. ICG imaging signals were collected both in NIR-I (850–950 nm) and NIR-II (1000–1700 nm) region under 808 nm excitation. Color bar ranges from 0 to 20,000 for BTC1070 and ICG, 0–5,000 for IR26. The detailed imaging parameters for each image are listed in Supplementary Table 1. **c** Measured signal-to-background ratio (SBR, defined in Supplementary Figure 8) of capillary images as a function of depth. By selecting a threshold SBR (SBR = 2, red dotted line) to define the level at which structure are minimally resolvable, the maximum penetration depth range for all fluorophores are showed as origin boxes in **b**. The bars represent mean ± s. d. derived from $n = 3$ line profiles measured at different positions in the capillary images. **d** Wavelength-dependent full-width at half-maximum (FWHM) of cross-sectional profiles in capillary images as a function of depth. The bars represent mean ± s.d. derived from the uncertainty in the Gaussian fitting of feature width. Source data are provided as a Source Data file

be beneficial for long-time observation such as imaging-guided surgical operation.

**pH-responsive properties of NIR-II BTCs.** The superior NIR-II bioimaging performance as well as the interesting photophysical properties of diethylamino-substituted BTCs inspired us to explore the possibility to develop fluorescent probes by studying their pH-responsive properties. Owing to the anti-quenching properties, these fluorophores are able to exhibit spectra responses to pH in aqueous solution (Fig. 5, Supplementary Figure 18, 19). For example, BTC1070 exhibits decrease of the absorbance band at 1015 nm with the appearance of a new broad band ~ 600–900 nm when downregulating the pH from 5 to 2 (Fig. 5b). And an intense peak at 950 nm is observed when pH further decreased to 0. The p$K$a values are calculated to be 0.29 and 3.81, clearly demonstrating the double protonation feature (Fig. 5c). In addition, ratiometric fluorescence response with large peak shift from 1065 to 980 nm when pH changed from 5 to 0 is observed under 808 nm excitation, accompanying with obvious intensity changes (Fig. 5d). Plot of integrated intensity ratio from two wavelength regions (1000–1300 nm and 900–1300 nm) at pH 0-7 is well fitted by a sigmoidal equation ($r^2 = 0.99$), suggesting an optimal pH-sensitive range is 1–4 (Fig. 5e).

We further performed $^1$H-NMR titration with deuterated TFA to study the protonation mechanism (Supplementary Figure 22). The downfield shifts of resonances of methylene protons on diethylamino groups reveal the stepwise protonation process on nitrogen atoms (Fig. 5a). Therefore, the complex absorption changes can be elucidated by the symmetry breaking and recovery, and the inhibition of ICT effect during the protonation process[29,34]. Specifically, the first protonation on one side strongly polarizes the polymethine chain, forming an unsymmetrical cyanine structure (BTC1070H$^+$) featuring hypsochromic and broaden absorption. Then the second protonation on another side regenerates a new symmetrical cyanine structure (BTC1070H$_2^{2+}$), resulting in intense absorption again. Owing to the inhibition of ICT effect, BTC1070H$_2^{2+}$ exhibits blue-

shifted absorption and fluorescence as well as significant enhanced fluorescence intensity.

**Ratiometric fluorescence imaging of pH in tissue phantom.** As a proof of concept, we further performed ratiometric fluorescence imaging of pH in vivo by using BTC1070 as a probe. Given that wavelength-dependent light attenuation in vivo will compromise the accuracy and reliability, we investigated the influence of tissue thickness on the performance of pH ratiometric sensing via phantom imaging. As shown in Fig. 6a, blurry capillary edges and significantly reduced fluorescent signals are observed from red channel (900–1700 nm) and cyan channel (1000–1700 nm) with increase of penetration depth, which can be attributed to the light attenuation and scattering. By contrast, ratiometric imaging clearly recognizes the capillaries against the background with consistent pseudo-colors for all pH groups (Fig. 6b). Furthermore, the ratios derived from different depths have the coefficient of variation lower than 4.12% for all pH groups, indicating high reliability of the ratiometric signals with respect to penetration depth (Supplementary Table 6). By analyzing the signal attenuation coefficients of red channel ($\tau_1$) and cyan channel ($\tau_2$), little difference ($\Delta\tau$) between them are observed for all pH groups (Fig. 6c), which indicates that the attenuation of fluorescence signals at these two channels are synchronous. Through a mathematic model shown in Supplementary Note 2, we confirm that a small $\Delta\tau$ ($|\Delta\tau| < 0.03$) is vital for the high reliability of the ratiometric signals (Supplementary Figure 23, Supplementary Table 6). Thus, even taking consideration of the absorption and scattering of tissue, ratiometric signals in different depths as a function of pH could be well calibrated by sigmoidal equations ($r^2 > 0.98$, Fig. 6d, Supplementary Note 3), in accordance with the fluorescence spectra results.

**In vivo ratiometric fluorescence imaging of gastric pH.** Based on the calibration results, we further tested the ability of BTC1070 for ratiometric imaging of gastric pH in vivo, as the gastric pH has a crucial role in maintaining the stomach's digestive function, preventing infection and affecting the availability of oral drugs[35].

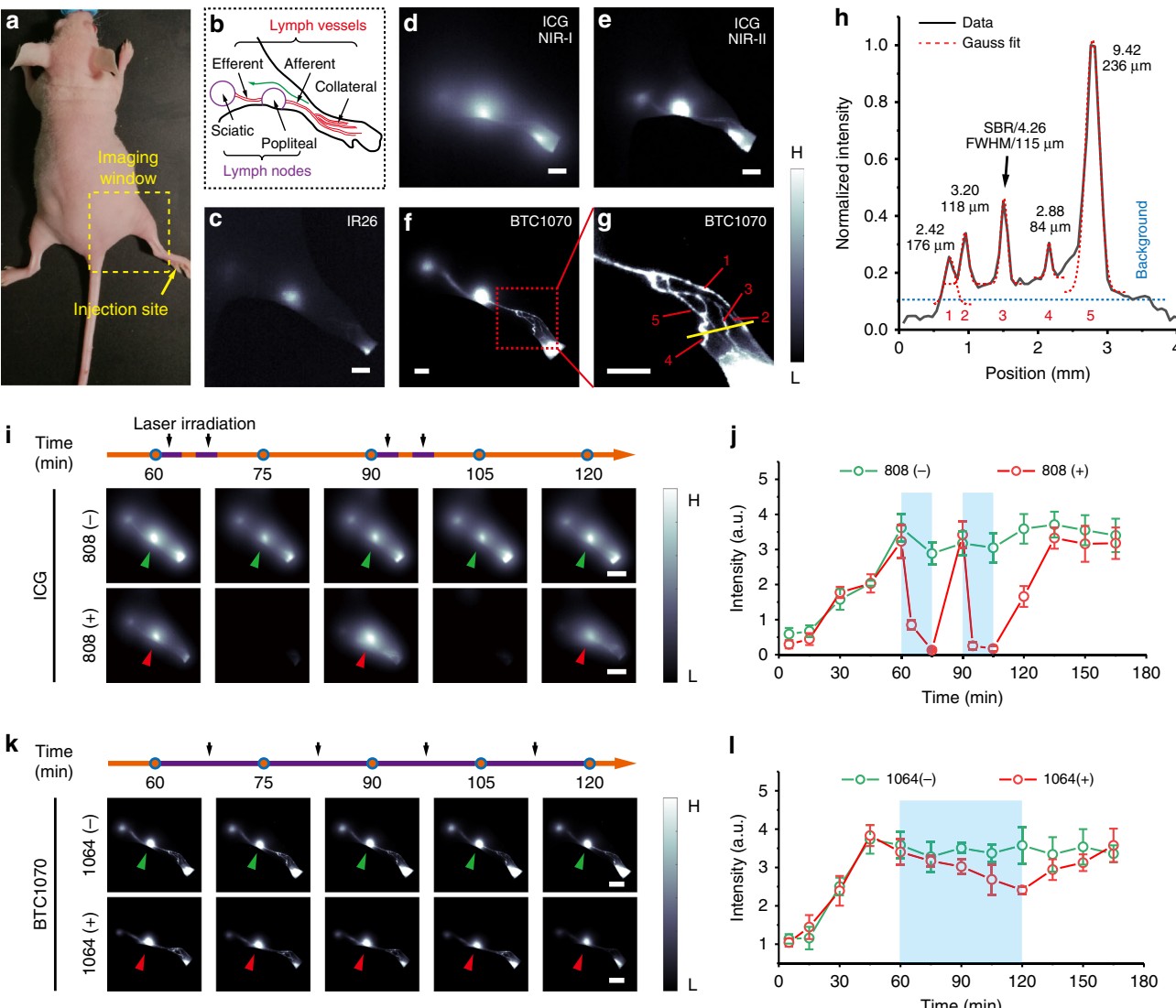

**Fig. 4** Superior lymphatic imaging with BTC1070 to ICG and IR26. **a** Digital photograph of a nude mouse fixed on imaging plate, showing the injection site (yellow arrow) of contrast agents and the lymphatic drainage imaging window (dash square). **b** Schematic illustration of the anatomical structure of lymphatic system in the hindlimb of nude mice, green arrow represents the lymphatic drainage from the paw to the sciatic lymph node. **c–g** Fluorescence images of lymphatic drainage using IR26 **c**, ICG **d**, **e**, and BTC1070 **f**, **g** as contrast agents in the hindlimb of nude mice on an InGaAs camera. Scale bar, 2.5 mm. IR26 and BTC1070 imaging signals were collected at wavelength of 1200–1700 nm under 1064 nm excitation. ICG was excited at 808 nm, and images were collected in the NIR-I (850–950 nm) and NIR-II (1000–1700 nm) region, respectively. **g** High-magnification (×3) image of the ankle (red square in **f**), showing at least five collateral lymph vessels were resolved. **h** Cross-sectional fluorescence intensity profiles (black solid) and Gaussian fit (red dotted) along the yellow bar in **g**. **i–l** In vivo photobleaching studies of ICG and BTC1070. **i**, **k** Fluorescence images of lymphatic drainage at different time points post-injection using ICG **i**/BTC1070 **k**. 808 ( + )/1064 ( + ) represents that the 808/1064 nm laser irradiation was conducted according to the set on the time line (purple segments on orange lines, each segment in **i** represents 3 min) with the same fluence rate ( ~ 150 mW cm$^{-2}$), whereas 808 ( − )/1064 ( − ) represents no laser irradiation was conducted. Scale bar, 5 mm. **j**, **l** Fluorescence intensity signals of popliteal lymph nodes (green arrows/red arrows in **i**, **k**) versus time. Blue rectangles represent time windows of laser irradiation. Color bar ranges from 1000 to 30,000 for **d–f,i**, and **k**, 500–5000 for **c**, 2000–20,000 for **g**. The detailed imaging parameters for each image are listed in Supplementary Table 1. The bars represent mean ± s.d. derived from $n = 3$ biologically independent mice. Source data are provided as a Source Data file

Mice were administrated simulated gastric fluid with different pH (pH 1.3 and pH 2.5) to mimic the pH environment of human stomach. Following gavage of BTC1070 micelle solution, the mice were anaesthetized and further imaged on an InGaAs camera under 808-nm excitation. As shown in Fig. 7a, noninvasive ratiometric imaging not only recognizes stomach profile from the left lateral aspect of the abdomen, at a tissue depth of ~ 2–4 mm (Supplementary Figure 24), but also differentiates the two pH environments with sharp pseudo-color contrast. Similar results also emerge in invasive ratiometric imaging of gastric fluid wrapped in a thin gastric wall ( ~ 1 mm depth, Fig. 7b) and imaging of exposed gastric fluid (0 mm depth, Fig. 7c). Furthermore, tissue thickness has negligible influence on the ratio values of each group (Supplementary Figure 25, 26). We then converted the ratios to pH values by means of the calibration curves (Supplementary Note 3). The results show good agreement with that measured by standard pH meter, and the differences between them (ΔpHs) are determined to be lower than 0.18 and 0.22 pH units for low (pH = 2.10 ± 0.11) and high pH groups (pH = 3.08 ± 0.27), respectively (Fig. 7d, Supplementary Table 7). By the

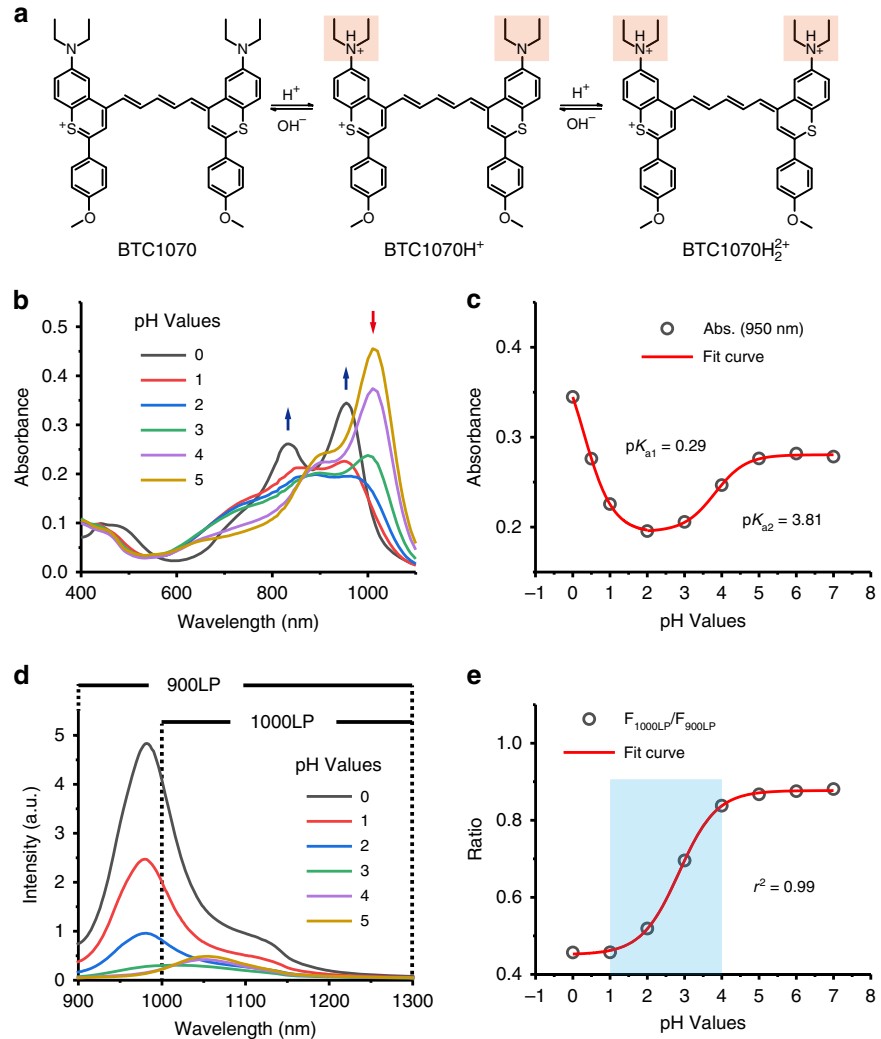

**Fig. 5** Protonation of BTC1070 and the corresponding spectra changes. **a** Protonation mechanism of BTC1070 derived from the [1]H-NMR titration in Supplementary Figure 22, showing stepwise protonation on nitrogen atoms. **b** Absorption spectra of 10 μM BTC1070 loaded into phospholipid nanomicelle in PBS at various pH values. Detailed data in a wider pH range of 0–9 are shown in Supplementary Figure 19. **c** Plot of absorbance at 950 nm versus pH values in the range of 0–7. The p$K_a$ values are calculated to be 0.29 and 3.81, respectively, based on Boltzmann curve fitting in origin software. **d** The corresponding fluorescence spectra excited at 808 nm at various pH values. **e** Plot of fluorescence ratio changes as a function of pH values. Ratio = $F_{1000LP}/F_{900LP}$, $F_{1000LP}$ and $F_{900LP}$ denote the integrated intensity at wavelength of 1000–1300 nm and 900–1300 nm, respectively

same method, gastric pH in normal mice (pH = 3.95 ± 0.18) can also be clearly visualized and accurately measured (ΔpHs ≤ 0.27, Supplementary Figure 27, 28). These results strongly suggest that BTC1070 is capable of noninvasively detecting gastric pH in a wide pH range by high-contrast deep-tissue ratiometric imaging.

## Discussion

In the present work, we focus on polymethine cyanine architectures because of their large absorption coefficient, tunable long-wavelength absorption/emission and facile modification[30]. So far, almost all cyanine fluorophores absorbing beyond 1000 nm were derived from heptamethine architectures. However, one severe limitation for biological applications has been the brightness quenching in aqueous solution induced by significant solvatochromism. Different from the commonly concept that aggregation causes fluorescence quenching in aqueous solution, the phenomenon of solvatochromism-caused quenching, unique for cyanine fluorophores, is significant even at a low dye-loading capacity of ~1 wt% in phospholipid micelle (Supplementary

Figure 5). Here, we demonstrate this major barrier could be effectively overcome by shortening polymethine chain length from 7 to 5 methine units. Thus, by further introducing electron-donating diethylamino moieties at different position of terminal heterocycles, stable absorption could be tuned from 800 to 1100 nm in aqueous solution, which is the key to obtain various anti-quenching properties. Significantly, these anti-quenching strategies could be further extended to various cyanines with other heterocycles for creating a library of bright and long-wavelength fluorophores. Moreover, enhanced brightness would be expected if heavy atoms or benzannelation can be eliminated from heterocyclic terminal groups[26], possibly opening new opportunities for in vivo imaging or sensing at the single-particle or single-cell level[36,37]. Nevertheless, shortening polymethine chain kills the original long wavelength as two methine units are equal to ~100 nm. Therefore, feasible strategies are still needed to push the wavelength limit of cyanine fluorophores in aqueous solution, and possible approaches might be focused on branching the polymethine chain or engineering the charge of π-system such as anionic or zwitterionic dyes[26,38,39]. Furthermore, another

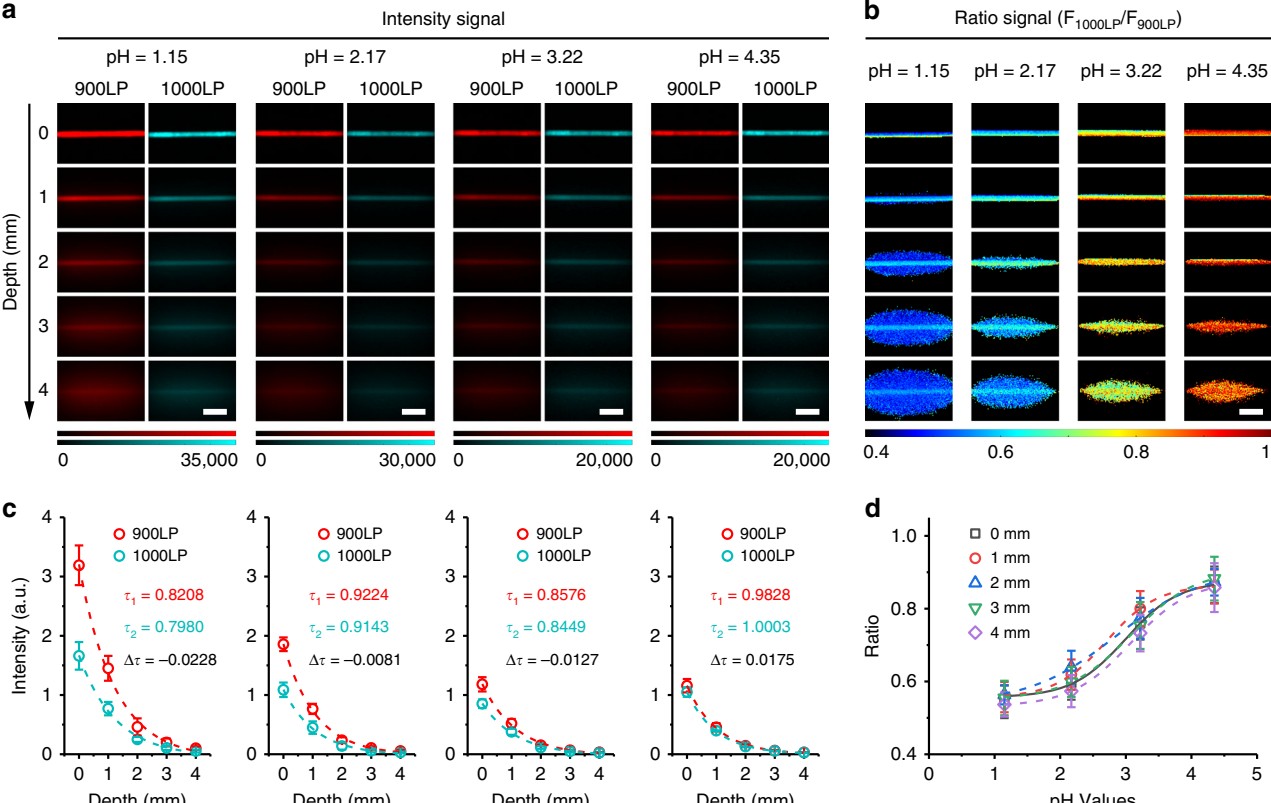

**Fig. 6** Ratiometric fluorescence imaging of pH with BTC1070 in mimic tissue. Fluorescence images **a** and corresponding ratiometric fluorescence images **b** of capillary tubes filled with BTC1070 micelle solution of different pH with varying depth from 0 to 4 mm. Excitation wavelength: 808 nm. Emitted signals were collected through passing 900LP ($F_{900LP}$, red channel) and 1000LP ($F_{1000LP}$, cyan channel) filters, respectively. $F_{1000LP}/F_{900LP}$ denotes the ratio signals. The detailed imaging parameters for each image are listed in Supplementary Table 1. Scale bar, 2.5 mm. **c** Measured intensity of capillaries as a function of depth (0–4 mm) in 1% Intralipid, corresponding to the fluorescence images of each pH value. $\tau_1$ and $\tau_2$ denote the signal attenuation coefficients of red and cyan channels, respectively, derived from the single-exponential fitting curves. The little differences between $\tau_1$ and $\tau_2$ ($\Delta\tau = \tau_1 - \tau_2$) are responsible for the reliable ratio signals with increased depth (Supplementary Note 2, Supplementary Figure 23). **d** Plots of measured ratio changes as a function of pH with varying depth. Curve fitting ($r^2 > 0.98$) is based on Boltzmann function in origin software (Supplementary Equation 2), generating the calibration functions at different depths (Supplementary Note 3). Data point with its error bar stands for mean ± s.d. derived from $n = 3$ replicated measurements of every pixel in the capillaries. Source data are provided as a Source Data file

unresolved but important issue is the functionalization and hydrophilicity of these cyanine fluorophores. In this study, except for the classical functionalization route on polymethine chain, the flexible synthetic methods for benzothiopyryliums such as buchwald-hartwing amination (Supplementary Figure 3) also enable facile functionalization either on the N- or O-substituent moieties. And through further PEGylation or antibody/peptide bioconjugation, the possibly resulting improvements on hydrophilicity, pharmacokinetics, targeting or retention will expand more biological applications for these fluorophores.

Near-infrared cyanines have always been of great interest in fluorescent biosensing application in vivo[30,40–43]. In this work, we explore the pH-responsive properties of diethylamino-substituted BTCs and shed light on the mechanism behind. Both BTC982 and BTC1070 show protonation occurred on two nitrogen atoms step by step, accompanying with significant absorption and fluorescence changes (Supplementary Figure 20–22). Through density functional theory calculation (Supplementary Figure 7), the average bond lengths of C–N connected to the π-system are calculated to be 137.3 pm for BTC982 and 138.4 pm for BTC1070. Both values are close to the intermediate between the typical C–N single (~ 147 pm) and double (~ 128 pm) bonds, indicating that there exists strong electronic delocalization between the nitrogen atoms and conjugated system. The delocalization makes the p$K$a

values of BTC982 (p$Ka_1$ = 0.72, p$Ka_2$ is too low to be estimated) and BTC1070 significantly lower than that of typical diethyla-nilin analogs (p$Ka$ ≈ 6.61) (Supplementary Figure 19). But despite low, replacing diethylamino with hydroxyl group may be a feasible strategy for designing physiological pH-responsive NIR-II probes, because phenolic hydroxyl group is also delocalizable and has a higher p$Ka$ (~ 10). In addition, owing to the delocalization effect, protonation on one side will lead to a significant breaking of donor strength of terminal groups, resulting in the blue-shift and broadening of absorbance spectra and reduced fluorescence for both BTC982 and BTC1070. Another interesting finding is ICT effect of nitrogen atoms at C-6 position; through the inhibition of this effect by protonation, BTC1070 show ratiometric fluorescence response with significant intensity change and wavelength shift. Overall, these discoveries may motivate the further research on the development of new NIR-II dyes and fluorescent sensors by modulating the π-electron system of benzothiopyrylium or other heterocycles.

In a proof of concept study, we used BTC1070 as a low-pH sensor for quantitative measurement of gastric pH, demonstrating the superior sensing accuracy of in vivo ratiometric imaging in the NIR-II region. Unlike in vitro ratiometric imaging with accurate and reliable quantification of live-cell events, to achieve the same accuracy in vivo remains impractical owing to the

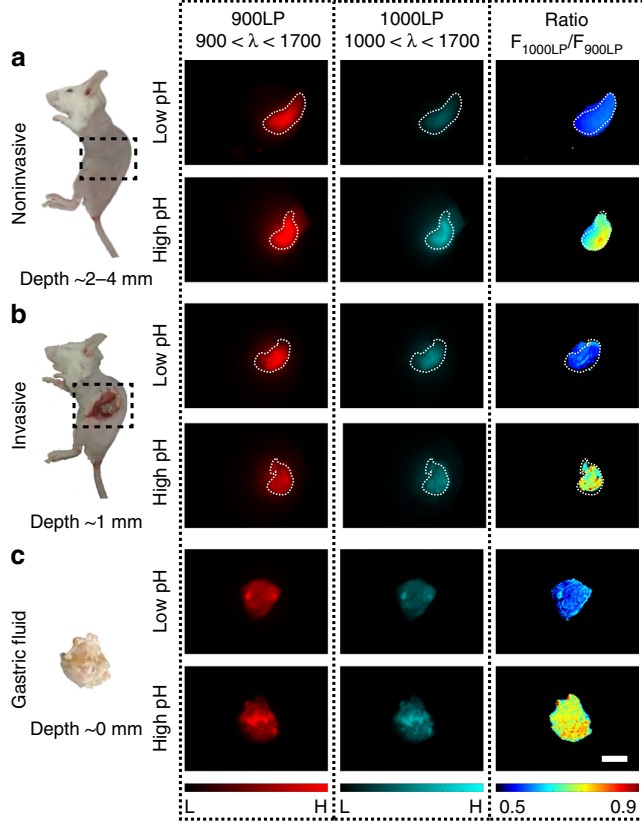

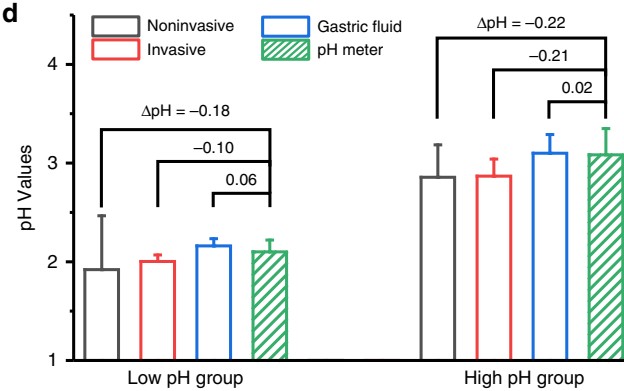

**Fig. 7** In vivo ratiometric fluorescence imaging of gastric pH. **a–c** Left: digital photographs of mice and dissected stomach denote three imaging modes. **a** noninvasive imaging at ~ 2–4 mm tissue depth; **b** invasive imaging of gastric fluid covered by ~ 1 mm thickness of gastric wall; **c** imaging of exposed gastric fluid. The measurement of tissue depth/thickness is shown in Supplementary Figure 24. Black dash square represents imaging window in mice. Right: fluorescence images and corresponding ratiometric fluorescence images of mice stomach at two different pH environments (red channel: 900–1700 nm; cyan channel: 1000–1700 nm; ratio channel: $F_{1000LP}/F_{900LP}$). Color bar ranges from 5000 to 30000 for the red/cyan channel in **a** and **b**, 0–30000 for the red/cyan channel in **c**. The detailed imaging parameters for each image are listed in Supplementary Table 1. Scale bar, 5 mm. **d**, Comparison of pH values in mice stomach measured by ratiometric fluorescence imaging and standard pH meter. The true gastric pH determined by pH meter is higher than that of simulated gastric fluid due to the buffering effect (see Methods). Ratios derived from the ratio images are converted to pH values by means of the calibration functions at corresponding depth (Supplementary Note 3). ΔpH = average pH resolved from ratiometric imaging−average pH measured by standard pH meter. Detailed pH data are summarized in Supplementary Table 7. Data point with its error bar stands for mean ± s.d. derived from $n = 3$ biologically independent mice. Source data are provided as a Source Data file

influence of wavelength-dependent light attenuation. Here, we demonstrate that this influence can be maximally reduced by picking adjacent and partially overlapped wavelength regions as response signal and reference signal because they show similar attenuation coefficients in tissue. Despite inevitable variations with increased depth, drawing calibration curves in mimic tissue imaging could further improve the reliability, as evidenced by quantifying gastric pH at a depth up to 4 mm. Therefore, the above strategies could be expected as a general methodology for in vivo quantitative or semi-quantitative ratiometric biosensing. Even so, there still exists great room for improvement in ratiometric imaging, especially for the image contrast. In Fig. 6b, increased pseudo-color backgrounds around the capillaries are observed with increased depth, which mainly derives from the ratiometric normalization of scattering signals in two channels. In biological tissue, tissue auto-fluorescence further increases the interference. To address this challenge, future efforts should be devoted to developing brighter and longer wavelength fluorophores or turning to the temporal domain for lifetime imaging[44], where long-lifetime NIR-II molecular fluorophores will be preferable.

In conclusion, a series of anti-quenching cyanine fluorophores based on pentamethine structure with stable peak absorption/emission up to 1015/1065 nm in aqueous solution were rationally designed and synthesized. The developed cyanine fluorophores exhibit enhanced brightness, photostability and imaging performance far superior to clinical-approved ICG, thus showing great potential for clinical translation. In addition, we have demonstrated the use of BTC1070 for noninvasive ratiometric quantification of gastric pH, showing reliable accuracy at a depth up to 4 mm comparable to standard pH electrode method. This study unlocks the potential of pentamethine cyanine fluorophores for NIR-II bioimaging and biosensing as well as other biological application.

## Methods

**General materials and instruments**. 1,2-dioleoyl-sn-glycero-3-phosphoethanolamine-*N*-[methoxy(polyethylene glycol)-2000] (ammonium salt) (DOPE-mPEG2000) was purchased from Avanti Polar Lipids. A total of 20% Intralipid emulsion was purchased from Sigma-Aldrich. Analytical grade solvents including petroleum ether, $CH_2Cl_2$, and ethyl acetate were purchased from Titan Scientific. All other chemicals were purchased from TCI and J&K without further purification unless otherwise noted. All $^1H$-NMR and $^{13}C$-NMR spectra were acquired on a Bruker AV-400 spectrometer. Chemicals shifts are referenced to the residue solvent peaks and given in ppm. MALDI-TOF MS analyses were performed in positive reflection mode on a 5800 proteomic analyzer (Applied Biosystems, Framingham, MA, USA) with a Nd: YAG laser. UV-Vis-NIR absorption spectra were recorded on Lamda750S PerkinElmer. NIR-II fluorescence spectra were recorded on Edinburgh Fluorescence Spectrometer FLS980 instrument with external 808 (MDL-III-2W), 915 (MDL-H-5W), or 1064 nm (MIL-III-1.5 W) semiconductor laser (Changchun New Industries Optoelectronics Tech. Co., Ltd.) as excitation source. In vivo NIR-II images were taken using NIRvana CCD camera (Princeton Instruments Inc.). Dynamic light scattering were carried out on a Malvern Zetasizer 3600 (Malvern Instruments).

**Synthesis of BTCs**. Synthesis methods for all compounds, characterization data are provided in the Supplementary Methods, Supplementary Figure 3, 29-52.

**General imaging setup**. The home-built imaging setup uses 808/915/1064-nm laser coupled into a 450-µm core metal-cladded multimode fiber (Changchun New Industries Optoelectronics Tech.) as illumination source. Fluorescence signal is directed from the imaging stage to the InGaAs SWIR camera (NIRvana 640, Princeton Instruments; 640 × 512 pixel) using a combination of various emission filters (Thorlabs and Edmund Optics) incorporated before camera lens (SWIR-35, Navitar). The whole assembly was surrounded by a partial enclosure to eliminate excess light while enabling manipulation of the field of view during operation. The InGaAs camera was cooled to − 80 °C, the analog to digital conversion rate was set to 2 or 10 MHz, the gain was set to high, and different exposure times were used to achieve sufficient signal. All images were background and blemish corrected within the LightField imaging software and processed with Matlab.

**Tissue phantom imaging study**. In all, 1% Intralipid® made by volumetric dilution of a commercial 20% stock solution was chosen as a mock tissue because of its similar scattering properties. Glass capillary tube ($\phi = 1$ mm) filled with sample

was wrapped in black tape to leave 2 cm length available for imaging. The capillaries were then placed under a cylindrical dish ($\phi = 50$ mm) and covered with different volumes of 1% Intralipid®. The depth of the capillary was calculated from the known area of the dish. The detailed imaging parameter sets (excitation lasers, fluence rates, emission filters, exposure times) were summarized in Supplementary Table 1–2. To obtain penetration depth information, average intensities were taken from the same region of interest at various depths and normalized according to the exposure times. To determine feature width, linear cross-sections were taken from the images and fit to Gaussians using Origin's built-in curve fitting. Ratio images were processed with Matlab software. Fitting the plot of average ratios derived from capillaries versus pH gave five calibration curves at various depths (0, 1, 2, 3, and 4 mm), and the corresponding calibration functions were listed in Supplementary Note 3, which were used for pH evaluation in vivo.

**Animal models**. All the following animal procedures were agreed with the guidelines of the Institutional Animal Care and Use Committee of Fudan University and performed in accordance with the institutional guidelines for animal handling. All of the animal experiments were permitted by the Shanghai Science and Technology Committee.

**In vivo fluorescence imaging of lymphatic drainage**. Female Balb/c nude mice (6 weeks old, average weight of 20 g) were purchased from Shanghai SLRC laboratory animal center. Before lymphatic imaging, all mice were anaesthetized using rodent ventilator with 2 L min$^{-1}$ air mixed with 4% isoflurane. All groups within study contained $n = 3$ mice. For visualization of lower limb collecting lymphatic vessels and drainage of rodent lymphatic system, contrast agents were injected intradermally into the dorsal skin of the rear paw. The injection dose was 50 μL PBS (pH 7.4) solution with varying concentration for BTC982 (1.25 nmol), BTC1070 (10 nmol), IR26 (10 nmol), and ICG (5 nmol). The detailed imaging parameter sets (excitation lasers, fluence rates, emission filters, and exposure times) were summarized in Supplementary Table 1. During the time course of imaging, the mouse was kept anaesthetized by a nose cone delivering air mixed with 4% isoflurane. All images were recorded at 60 min post-injection to visualize the drainage from lower limb to upper limb through flank trunk.

**In vivo photobleaching study**. For photobleaching study, mice in control group without laser irradiation were firstly imaged at 5, 15, 30, 45, 60, 75, 90, 105, 120, 135, 150, and 165 min post injection to visualize the dynamic drainage process. Constant fluorescence signals of lymph nodes were observed at the time range between 60- and 120-min post injection. Then, the mice hindlimbs of another group were intermittently irradiated (3 min for each time, time interval of 2 min) at the same time range (60–120 min post injection) using another laser with same wavelength and larger fluence rate ( ~ 150 mW cm$^{-2}$) for fluorescence bleaching. 1064-nm laser irradiation was conducted continuously because of its minimum heating effect in biological tissue. The total irradiation times were 6 min for ICG (3 min for complete photobleaching), 27 min for BTC982 and 60 min for BTC1070. The imaging time intervals of mice with laser irradiation were same as that in control group without laser irradiation. For ICG, two time points of 65 and 95 min were added due to the fast photobleaching after 3 min laser irradiation. The results of BTC982 are shown in Supplementary Figure 15–16.

**In vivo ratiometric fluorescence imaging of gastric pH**. Female ICR mice (8 weeks old, 26 ~ 28 g) purchased from Shanghai SLRC laboratory animal center were shaved and fasted for 12 h with free access to water prior to the experiment. The gastric pH of fasted mice before oral gavage was measured to be $3.83 \pm 0.2$ ($n = 6$) using a pre-calibrated pH meter (Seven Easy, Mettler-Toledo, Switzerland) equipped with a micro electrode (Inlab®Micro, 3 mm shaft diameter), indicating the gastric pH of normal mice is still capable of being measured with BTC1070. Subsequently, mice were randomly divided into three groups ($n = 3$), where two groups were gavaged orally with 200 μL simulated gastric fluid of different pHs (pH 1.3 and 2.5) to simulate the pH environment in human stomach (pH 1–3), and the other group was used as normal control group. After that, all mice were administrated with 20 μL concentrated BTC1070 micelle solution (500 μM dye in deionized water) and anaesthetized using rodent ventilator with 2 L min$^{-1}$ air mixed with 4% isoflurane. Noninvasive imaging was carried out at the left lateral aspect of mice abdomen. For invasive imaging, mice were sacrificed and dissected along the upper part of the abdomen. The stomachs were exposed outside from the abdomen for imaging. Then the stomachs were excised and the gastric pH was immediately measured by fully immersing the electrode tip in the stomach lumen. The true gastric pH determined by pH meter was higher due to the buffering effect. For gastric fluid imaging, the excised stomachs were cut open along the greater curvature, excess gastric content was removed. Imaging was taken on the inner wall of stomachs. The excitation source was provided by an 808 nm diode laser with fluence rate of ~ 200 mW cm$^{-2}$, and the emission signals were collected at 900–1700 nm and 1000–1700 nm with same exposure time of 50 ms. Ratio images were processed with Matlab software. Average ratios were taken from the same region of interest at various images. pH values were resolved by means of the calibration function shown in Supplementary Note 3.

**Reporting summary**. Further information on experimental design is available in the Nature Research Reporting Summary linked to this article.

## Data availability
The data that support the findings of this study are available from the corresponding authors upon reasonable request. The data underlying Figs. 1k, 3c,d, 4j,1, 6c, 7d, Supplementary Figures 9, 10b-d, 11b-d, 14f, g, 15, 16b, 23, 27–28 are available in the Source Data file.

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

## Acknowledgements

This work was supported by the National Key R & D Program of China (2017YFA0207303), National Natural Science Fund for Distinguished Young Scholars (21725502), Key Basic Research Program of Science and Technology Commission of Shanghai Municipality (17JC1400100).

## Author contributions

F.Z. and S.W. designed the project. S.W. and D.L. synthesized and characterized the compounds. Z.L. and T.W contributed to the spectra characterization. Y.F. and L.L. built the optical system and contributed to the imaging processing with Matlab software. C.S. and S.W. conducted the animal experiments. S.W. was primarily responsible for data collection. S.W. and F.Z. analyzed the results, prepared the manuscript, Figures and supplementary information. All authors contributed to discussion and editing of the manuscript.

## Additional information

**Competing interests:** The authors declare no competing interests.

