## [Peer Review File · Nature Communications]

Reviewers' comments:

Reviewer #1 (Remarks to the Author):

Wang et. al. synthesized a series anti-quenching pentamethine cyanine fluorophores that demonstrate a number of improvements over current NIR-II small molecule imaging agents. As cyanine-based dyes with NIR-II emission show large spectral changes as a function of solvent polarity, the authors shortened the polymethine chain length and systematically altered electron donors. Impressively, the absorbance of the resulting BTC dyes showed only minor attenuation in water compared to commercially available NIR-II dyes including IR26 where absorbance drops by ~98% even when encapsulated in micelles. BTC's reduced quenching in water results in improved NIR-II optical properties compared to both ICG and IR26, demonstrated through both phantom and mouse studies. Even at a physiological pH where BTC1070's NIR intensity should be reduced compared to acidic conditions, the dyes produce high quality lymphatic images. Interestingly, BTC1070's optical properties vary as function of pH through nitrogen atom protonation. Changes in BTC1070's absorbance/fluorescence spectral power distribution enable ratiometric imaging between pH 1-4. The manuscript proceeds to show that dividing NIR-II sub-regions can resolve simulated gastric pH changes in mouse models.

The manuscript highlights a new strategy to reduce water quenching of NIR-II organic dyes. This is the first example of a pH biosensing NIR-II probe which results in impressive stomach imaging without significant feature broadening. The authors employ an innovative method to leverage these results for single-dye ratiometric imaging. The manuscript is well constructed, results are mostly supported by strong experimental data, and conceptually novel for the NIR-II range. However, the following major revisions should be made along with the collection of supporting experimental data prior to acceptance.

- Significant spelling and grammatical errors throughout the manuscript including 'continues 808 nm laser' on page 4 and 'mehine' on page 8.
- Most fluorescent images do not contain intensity scale bars such as Figure 3b. Additionally, Figure captions should either list imaging parameters such as exposure time, filter sets, etc. or direct the reader to the SI where all of this information is listed.
- In Figure 3d, why do ICG, BTC1070, and IR26 all show different FWHMs as a function of imaging depth? It seems that all dyes in the same imaging window should have similar FWHM. These differences may be due to intensity differences as opposed to scattering behavior. Information on if/how these capillary tubes were normalized should be provided.
- In Figure 4d-g, there are no intensity scale bars on any of these images. Also, it seems that intensity thresholding on all these images is different based on relative popliteal LN sizes, preferentially highlighting features in 4f and 4g. An explanation of image normalization methodology should be provided.
- Lymphatic circulation differences in Figure 4 f,g are likely due to inter-animal model variability as opposed to fundamental fluorophore differences. There is a high degree of variability in lymphatic circulation vasculature in mouse hindlimbs. The number of mice used should be noted and images from other mice experiments provided in the SI to truly show lymphatic visualization differences resulting from improved BTC1070's optical properties.
- Figure 4d and 4e say NIR-I and NIR-II, but the imaging settings in Table S1 list both images as taken with 850 LP/950 SP (i.e., NIR-I). This serious typo needs to be addressed. Further, all these imaging parameters need to be presented in the manuscript and easily accessible for comparisons. At minimum, the SI table should be referenced in the figure caption.
- If the NIR-II optical properties of BTC1070 are truly superior to ICG's NIR-II emission, why does BTC1070 require a 2-fold higher injection dose and a 4-fold higher exposure time between Figure 4e and g? This is completely opposite to Figure 1e. The typo on Table S1 makes it extremely hard to make this type of comparison.
- Table S1 needs to be re-written for EACH image in the manuscript. Grouping many Figure panels together along with ~5 filters in each row makes it almost impossible to understand the setup

imaging parameters in each Figure/panel.

- Labeling in Figure 1b makes it impossible to understand the different Figure panels. There are 8 graphs/chemical structures that are all grouped under Figure 1b. Also, the background of the graph with the 'ideal state' chemical drawing is for some reason grey??
- Figure 1f should be done in PBS and/or FBS, not DMSO. The photostability in biological media is much more relevant for biomedical imaging.
- Why does the emission spectrum of BTC1070 in Figure 5d not have a dip ~1150 nm while the emission spectrum in Figure 1d has a prominent dip ~1150 nm?
- A plot of the integrated NIR-II emission should be plotted as a function of pH. Based on Figure 3a's intensity scale and Figure 5d's emission spectrum, it seems that at physiological pH's the NIR-II intensity will be significantly lower than in acidic conditions. This probably explains why high 200 ms exposure times are needed for mouse hind-limb lymphatic imaging.
- The quantum yield in Table 1 needs to be listed in PBS in addition to DCM. Most other published NIR-II imaging agent quantum yields are listed in PBS.
- From a quick literature search, the pH of mouse stomachs is ~4 which is still capable of being measured with BTC1070. This should be done as a control to show differences relative to simulated stomach acid.
- In the Discussion section, the authors should list possible steps to synthesize NIR-II pH sensitive probes that are sensitive near physiological pH values for ratiometric imaging.
- In the Discussion section, the authors should discuss possible steps to eliminate the need for micelle formulations. Micelle formulations hinder rapid excretion pharmacokinetics and are unlikely to yield renal clearance.

Reviewer #2 (Remarks to the Author):

This paper details the synthesis of NIR II fluorescent cyanines for biological imaging applications. The authors utilize a phenyl substituted benzothioopyrylium heterocycle to red-shift pentamethine cyanines into the second NIR window and compare the efficacy of these dyes versus the standard ICG using both the NIR I and NIR II windows for ICG.

The paper presents a wealth of experimental data for the new compounds; however there are major flaws which must be corrected before the manuscript can be accepted for publication.

First, the English language throughout needs thorough proof-reading. The manuscript is very difficult to follow at many points.

Throughout the manuscript, current NIR II dyes are described as "quenching absorbance" in polar solvents; quenching specifically refers to the reduction in fluorescence signal. The absorbance of these dyes broadens and undergoes a hypsochromic shift. Additionally, no mention of aggregation or other sources of this behavior are mentioned despite playing a significant role in the poor optical properties of cyanine dyes in aqueous solutions.

In the synthetic scheme, compounds 2 & 3c should have the diethylamino substituent, not the bromo group.

In the absorbance spectra, all of the synthesized dyes have a small shoulder peak at ~1150 nm. This needs to be discussed in the text.

For the computational modeling, BTC 982 shows a higher torsional angle between the two heterocycles; however, the discussion fails to postulate why this would be the case. Some discussion needs to be directed at explaining why this might be the case.

The NMR titrations for the pKa studies are confusing. The spectra for the two compounds do not

appear to be substantially different; however, the authors conclude that one dye is protonating on the polymethine chain, then, upon reaching the second pKa that proton is transferred to the diethylamino group as the other amine protonates as well. This seems poorly supported by the titration studies and is rather difficult to see as being the case.

Point-to-Point Response to Reviewers' Comments

We greatly appreciate the very valuable and constructive comments from the reviewers which give us the opportunities to improve this manuscript. This manuscript has been revised accordingly. Additional details and many new results have been included. Below are our responses to reviewers' comments.

Reviewer #1 (Remarks to the Author):

Wang et. al. synthesized a series anti-quenching pentamethine cyanine fluorophores that demonstrate a number of improvements over current NIR-II small molecule imaging agents. As cyanine-based dyes with NIR-II emission show large spectral changes as a function of solvent polarity, the authors shortened the polymethine chain length and systematically altered electron donors. Impressively, the absorbance of the resulting BTC dyes showed only minor attenuation in water compared to commercially available NIR-II dyes including IR26 where absorbance drops by ~98% even when encapsulated in micelles. BTC's reduced quenching in water results in improved NIR-II optical properties compared to both ICG and IR26, demonstrated through both phantom and mouse studies. Even at a physiological pH where BTC1070's NIR intensity should be reduced compared to acidic conditions, the dyes produce high quality lymphatic images. Interestingly, BTC1070's optical properties vary as function of pH through nitrogen atom protonation. Changes in BTC1070's absorbance/fluorescence spectral power distribution enable ratiometric imaging between pH 1-4. The manuscript proceeds to show that dividing NIR-II sub-regions can resolve simulated gastric pH changes in mouse models.

The manuscript highlights a new strategy to reduce water quenching of NIR-II organic dyes. This is the first example of a pH biosensing NIR-II probe which results in impressive stomach imaging without significant feature broadening. The authors employ an innovative method to leverage these results for single-dye ratiometric imaging. The manuscript is well constructed, results are mostly supported by strong experimental data, and conceptually novel for the NIR-II range. However, the following major revisions should be made along with the collection of supporting experimental data prior to acceptance.

Comments

1. **Significant spelling and grammatical errors throughout the manuscript including 'continues 808 nm laser' on page 4 and 'mehine' on page 8.**

Response: Thanks so much for the reviewer's comments. We have polished our manuscript carefully and corrected the grammatical, styling, and typos found in our manuscript. Detailed modifications are also provided as following:

Page 3, line 33 & Page 4, line 2: "*showed*" was changed to "*show*".

Page 4, line 3: "*continues 808 nm laser*" was changed to "*continues-wave 808 nm laser*".

Page 7, line 29: "*with a tissue coverage depth of ~2-4 mm*" was changed to "*at a tissue depth of ~2-4 mm*".

Page 8, line 18: "*mehine*" was changed to "*methine*".

Page 8, line 22: "*other donor end-groups*" was changed to "*various cyanines with other heterocycles*".

Page 8, line 29: "*the charged π -system*" was changed to "*the charge of π -system*".

Additionally, we have highlighted other modifications in the revised manuscript.

2. Most fluorescent images do not contain intensity scale bars such as Figure 3b. Additionally, Figure captions should either list imaging parameters such as exposure time, filter sets, etc. or direct the reader to the SI where all of this information is listed.

Response: We have added intensity scale bars in Figure 3b and Figure 4d-g. The intensity scale range is listed for each fluorescent image in the revised figure captions. Additionally, we also referenced Supplementary Table 1 in these figure captions, which contains all of the imaging parameters.

We have highlighted all changes in the revised figure caption such as:

Figure 1k: "*Color bar ranges from 0 to 30,000 for each image. Excitation wavelength: 808 nm/~100 mW cm⁻² for ICG, 915 nm/~100 mW cm⁻² for IR26 and BTCs. The detailed imaging parameters for each image are listed in Supplementary Table 1.*"

Figure 3: "*Color bar ranges from 0 to 20,000 for BTC1070 and ICG, 0 to 5,000 for IR26. The detailed imaging parameters for each image are listed in Supplementary Table 1.*"

Figure 4: "*Color bar ranges from 1,000 to 30,000 for d-f, i and k, 500 to 5,000 for c, 2,000 to 20,000 for g. The detailed imaging parameters for each image are listed in Supplementary Table 1.*"

Figure 6: "*The detailed imaging parameters for each image are listed in Supplementary Table 1.*"

Figure 7, Supplementary Figure 25-27: "*Color bar ranges from 5000 to 30000 for the red/cyan channel in a and b, 0 to 30000 for the red/cyan channel in c. The detailed imaging parameters for each image are listed in Supplementary Table 1.*"

Supplementary Figure 8-9: "*Color bar ranges from 0 to 40000 for 1100LP channel, 0 to 20000 for 1200LP channel. The detailed imaging parameters for each image are listed in Supplementary Table 1.*"

Supplementary Figure 14: "*Color bar ranges from 1,000 to 30,000. The detailed imaging param-*

eters for each image are listed in Supplementary Table 1.”

Supplementary Figure 15: “Color bar ranges from 1,000 to 30,000 for a and d, 2,000 to 20,000 for b and e. Images were taken at wavelength of 1200-1700 nm under 1064 nm excitation. The detailed imaging parameters for each image are listed in Supplementary Table 1.”

3. In Figure 3d, why do ICG, BTC1070, and IR26 all show different FWHMs as a function of imaging depth? It seems that all dyes in the same imaging window should have similar FWHM. These differences may be due to intensity differences as opposed to scattering behavior. Information on if/how these capillary tubes were normalized should be provided.

Response: In Figure 3d, the following contents were listed in the figure caption:

“BTC1070 and IR26 imaging signals were collected in 1200-1700 nm region under 1064 nm excitation. ICG imaging signals were collected both in NIR-I (850-950 nm) and NIR-II (1000-1700 nm) region under 808 nm excitation.”

As photon scattering decreases at progressively longer wavelength, ICG imaging signals collected in NIR-I (850-950 nm) region has the lowest resolution (broad FWHMs), while BTC1070 imaging signals collected in 1200-1700 nm region shows the highest resolution (narrow FWHMs). However, BTC1070 shows narrower FWHMs than IR26 despite imaging at the same excitation and emission wavelength. This phenomenon should be attributed to the higher signal-to-background ratio (SBR) of BTC1070 due to its higher brightness.

To make this point clearer for the readers, we have added the following Supplementary Figure 6 in the revised Supporting Information, clearly showing the information of cross-sectional intensity profiles.

Continued on Next Page

Supplementary Figure 6. Cross-sectional intensity profiles of capillary images in Figure 3b at various imaging depths. **a**, BTC1070, **b**, IR26, **c**, ICG (NIR-I, 850-950 nm), **d**, ICG (NIR-II, 1000-1700 nm). FWHMs are derived from the curve fitting using Gaussian function in origin software. SBRs are calculated by dividing average signal intensity of capillary (6 pixels width) by the average background signal intensity (6 pixels width, 14 pixels apart from the capillary signal).

4. In Figure 4d-g, there are no intensity scale bars on any of these images. Also, it seems that intensity thresholding on all these images is different based on relative popliteal LN sizes, preferentially highlighting features in 4f and 4g. An explanation of image normalization methodology should be provided.

Response: In the revised manuscript, we have added intensity scale bar for Figure 4c-g, and the intensity scale range are listed for each figure in the figure caption (please see our response to Question #2 in “Comments”).

In our originally submitted manuscript, color bar ranges from 1,000 to 30,000 for Figure 4d, e and k, 500 to 5,000 for Figure 4c, 1,000 to 20,000 for Figure 4f and g. In order to show the collateral lymph vessels clearly, we set a relatively low intensity thresholding (1,000 to 20,000) for Figure 4f, because their signals are much lower than the popliteal lymph node signal. As a result, the popliteal lymph node size of Figure 4f is relatively larger than other groups because all signals beyond 20,000 are normalized to 20,000. Nevertheless, to further highlight the collateral lymph vessel features, we focused on the ankle in the mice hindlimb and acquired images as show in Figure 4g, through changing some imaging parameters such as laser irradiation position, angle and power. As a result, **the different imaging parameters caused the difference of the highlighting features in Figure 4f and 4g.**

In the revised Manuscript, we have redesigned the lymphatic drainage imaging (including the photostability experiment of Figure 4k and 4l) by using BTC1070 as contrast agent, where the imaging parameters for Figure 4g are same as Figure 4f (see Supplementary Table 1 in our response to Question #8 in “comments”). The intensity thresholding (1,000 to 30,000) for Figure 4f (BTC1070) is set to the same range as Figure 4d and e (ICG). Then to highlight the fine structure of interlaced collateral lymph vessels, we zoomed in the ankle (red dashed square in Figure 4f) and showed the image in Figure 4g at 3× magnification with an intensity thresholding of 2,000 to 20,000.

The Figure 4f, g, k and l are added to the new Figure 4 in the revised manuscript as follows:

Figure 4 | Superior lymphatic imaging with BTC1070 to ICG and IR26. **a**, Digital photograph of a nude mouse fixed on imaging plate, showing the injection site (yellow arrow) of contrast agents and the lymphatic drainage imaging window (dash square). **b**, Schematic illustration of the anatomical structure of lymphatic system in the hindlimb of nude mice, green arrow represents the lymphatic drainage from the paw to the sciatic lymph node. **c-g**, Fluorescence images of lymphatic drainage using IR26 (**c**), ICG (**d, e**) and BTC1070 (**f, g**) as contrast agents in the hindlimb of nude mice on an InGaAs camera. Scale bar, 2.5 mm. IR26 and BTC1070 imaging signals were collected at wavelength of 1200-1700 nm under 1064 nm excitation. ICG was excited at 808 nm, and images were collected in the NIR-I (850-950 nm) and NIR-II (1000-1700 nm) region, respectively. **g**, High-magnification (3 \times) image of the ankle (red square in **f**), showing at least 5 collateral lymph vessels be resolved. **h**, Cross-sectional fluorescence intensity profiles (black solid) and Gaussian fit (red dotted) along the yellow bar in **g**. **i-l**, *In vivo* photobleaching studies of ICG and BTC1070. **i, k**, Fluorescence images of lymphatic drainage at different time points post-injection using ICG (**i**)/BTC1070 (**k**) with (+)/without (-) laser irradiation. Laser irradiation was conducted according to the set on the time line (purple region, each segment in **i** represents 3 min), and the irradiation fluence rates were all set to ~ 150 mW cm $^{-2}$. **j, l**, Fluorescence intensity signal of popliteal lymph node (green arrows/red arrows in **i, k**) with (+)/without (-) laser irradiation versus time. Blue region: time window of laser irradiation. **Color bar** ranges from 1,000 to 30,000 for **d-f, i** and **k**, 500 to 5,000 for **c**, 2,000 to 20,000 for **g**. The detailed imaging parameters for each image are listed in Supplementary Table 1. The bars represent mean \pm s.d. derived from $n = 3$ biologically independent mice.

5. *Lymphatic circulation differences in Figure 4 f, g are likely due to inter-animal model variability as opposed to fundamental fluorophore differences. There is a high degree of variability in lymphatic circulation vasculature in mouse hindlimbs. The number of mice used should be noted and images from other mice experiments provided in the SI to truly show lymphatic visualization differences resulting from improved BTC1070's optical properties.*

Response: Thanks very much for the useful suggestion. **Imaging was performed independently on n = 3 biological replicates with similar result. We have provided two fluorescent images of lymphatic drainage in another two independent mice in the revised Supporting Information.** As shown in Supplementary Figure 15, both the images clearly show anatomical structure of lymphatic drainage in the hindlimb. Especially, 4-6 collateral lymph vessels are resolved from the magnified images with high feature resolution and high SBR, clearly demonstrating the superior imaging performance of BTC1070.

Supplementary Figure 15. Fluorescence images (a,d) of lymphatic drainage on a group of nude mice (Mouse 2 and 3) using BTC1070 as contrast agent. Scale bar, 2.5 mm. Color bar ranges from 1,000 to 30,000 for a and d, 2,000 to 20,000 for b and e. Images were taken at wavelength of 1200-1700 nm under 1064 nm excitation. The detailed imaging parameters for each image are listed in Supplementary Table 1. b,e, high-magnification (3×) images of the ankle (yellow square in a,d), showing 4-6 collateral lymph vessels be resolved. c,f, Cross-sectional fluorescence intensity profiles (black solid) and Gaussian fit (red dotted) along the yellow bar in b,e, respectively.

6. *Figure 4d and 4e say NIR-I and NIR-II, but the imaging settings in Table S1 list both images as taken with 850 LP/950 SP (i.e., NIR-I). This serious typo needs to be addressed. Further, all these imaging parameters need to be presented in the manuscript and easily accessible for comparisons. At minimum, the SI Table should be referenced in the figure caption.*

Response: Thanks so much for the useful suggestion. We have corrected the typo in the revised Supplementary Table 1. And the revised Supplementary Table 1 has been referenced in the figure caption, which lists all the imaging parameters.

7. *If the NIR-II optical properties of BTC1070 are truly superior to ICG's NIR-II emission, why does BTC1070 require a 2-fold higher injection dose and a 4-fold higher exposure time between Figure 4e and g? This is completely opposite to Figure 1e. The typo on Table S1 makes it extremely hard to make this type of comparison.*

Response: In Figure 1e, we collected the emission of ICG, IR26 and BTCs at 1000-1700 nm region for the NIR-II brightness comparison. Therefore, to eliminate the interference of excited light whilst ensure all dyes be effectively excited, ICG was excited at 808 nm, IR26 and BTCs were excited at 915 nm. This information has been added in the revised figure caption as following:

Figure 1k. *“NIR-II (1000-1700 nm) brightness comparison of equimolar (10 μ M) BTC980, BTC982, BTC1070, IR26 and ICG in PBS (pH 7.4) on an InGaAs camera. Inset: fluorescence images of capillaries filled with corresponding fluorophores. Color bar ranges from 0 to 30,000 for each image. Excitation wavelength: 808 nm/ \sim 100 mW cm⁻² for ICG, 915 nm/ \sim 100 mW cm⁻² for IR26 and BTCs. The detailed imaging parameters for each image are listed in Supplementary Table 1.”*

However, BTC1070 has a maximum absorption at 1015 nm, allowing for imaging under longer-wavelength excitation such as 1064 nm, which gives a deeper tissue penetration (Supplementary Figure 10 in the revised Supporting Information). Thus, the following bioimaging experiment using BTC1070 was carried under 1064 nm excitation, and images were collected with 1100 nm and 1200 nm long pass filters to remove the backscattered excitation light. **It should be noted that the photons emitted at 1200-1700 nm for BTC1070 only account for 8.1% of the total photons emitted in the NIR-II (1000-1700) spectra region. This leads to a high dose and a high exposure time to compensate for the reduced brightness.** Nevertheless, our tissue phantom study has demonstrated BTC1070 excited at 1064 nm and collected beyond 1200 nm has the deepest penetration depth, highest SBR and resolution among all dyes with various imaging wavelengths, which is benefited from the reduced light attenuation and scattering at longer wavelength. **Furthermore, the difference between ICG and BTC1070 micelle on other factors such as molecule/particle size, formulation and surface properties, which have been demonstrated to be vital in lymphatic drainage** (for example, *Nature* **2009**, 462, 449-460; *ACS Nano* **2017**, 11, 4247-4255; *ACS Nano* **2013**, 7, 8645-8657), **may also bring the difference of imaging parameters for *in vivo* and *in vitro* imaging.**

8. *Table S1 needs to be re-written for EACH image in the manuscript. Grouping many Figure panels together along with ~5 filters in each row makes it almost impossible to understand the setup imaging parameters in each Figure/panel.*

Response: We appreciate the reviewer for the useful suggestion. In the revised Supporting Information, we have re-written Supplementary Table 1 as follow.

Continued on Next Page

Supplementary Table 1. Details of imaging parameters used in each fluorescent image of this study.

Figure	Dyes	Concentration/Dose	Laser /Fluence rate (nm/mWcm ⁻²)	Filter sets	Exposure time (ms)
Fig. 1k (inset)	BTCs, IR26	10 μ M	915/~100	850LP, 1000LP	20
Fig. 1k (inset)	ICG	10 μ M	808/~100	850LP, 1000LP	20
Fig. 3b	BTC1070, IR26	200 μ M	1064/~100	1100LP, 1200LP	See Supplementary Table 2
Fig. 3b	ICG (NIR-I)	100 μ M	808/~100	850LP, 950SP	
Fig. 3b	ICG (NIR-II)	100 μ M	808/~100	850LP, 1000LP	
Fig. 4c	IR26	10 nmol	1064/~60	1100LP, 1200LP	200
Fig. 4d, 4i	ICG	5 nmol	808/~10	850LP, 950SP	50
Fig. 4e	ICG	5 nmol	808/~10	850LP, 1000LP	50
Fig. 4f, 4g, 4k, Supplementary Fig. 15	BTC1070	10 nmol	1064/~60	1100LP, 1200LP	200
Fig. 6a (red channel)	BTC1070	200 μ M	808/~250	900LP	See Supplementary Table 2
Fig. 6a (cyan channel)	BTC1070	200 μ M	808/~250	900LP, 1000LP	
Fig. 7a-c, Supplementary Fig. 25-27 (red channel)	BTC1070	10 nmol	808/~200	900LP	50
Fig. 7a-c, Supplementary Fig. 25-27 (cyan channel)	BTC1070	10 nmol	808/~200	900LP, 1000LP	50
Supplementary Fig. 8 (1100LP)	BTC1070	200 μ M	1064/~100	1100LP	See Supplementary Table 2
Supplementary Fig. 8 (1200LP)	BTC1070	200 μ M	1064/~100	1100LP, 1200LP	
Supplementary Fig. 9 (1100LP)	BTC1070	200 μ M	915/~100	1000LP, 1100LP	
Supplementary Fig. 9 (1200LP)	BTC1070	200 μ M	915/~100	1000LP, 1100LP, 1200LP	
Supplementary Fig. 13	BTC982	1.25 nmol	915/~4	850LP, 1000LP	5, 25, 50, 100
Supplementary Fig. 14	BTC982	1.25 nmol	915/~4	850LP, 1000LP	50

9. Labeling in Figure 1b makes it impossible to understand the different Figure panels. There are 8 graphs/chemical structures that are all grouped under Figure 1b. Also, the background of the graph with the ‘ideal state’ chemical drawing is for some reason grey??

Response: Thanks very much for the useful suggestion. We have labelled the 4 chemical structures as Figure 1b, d, e, f, and the 4 graphs as Figure 1c, g, h, i and revised the figure caption of Figure 1 (highlight in the manuscript). Additionally, we simplified Figure 1c (previous Figure 1b) for easier understanding, including the removal of gray background and other redundant lines.

We retained the “*ideal state*” and “*quenching state*” chemical drawings in Figure 1c, because they are helpful to understand the basic mechanism of solvatochromism-caused quenching for NIR-II long-chain cyanines. In the revised figure caption of Figure 1, we used the following revised contents (highlight in Figure 1c) to explain these chemical drawings:

Figure 1c. “*The intense and narrow absorption in DCM corresponds to a symmetric cyanine featuring equalized charge and minimal bond length alternation, which is the “ideal state” called as cyanine limit. Polar solvents like DMSO, H₂O break the symmetry, leading to weak and broad absorption, corresponding to a “quenching state” called as beyond the cyanine limit.*”

10. Figure 1f should be done in PBS and/or FBS, not DMSO. The photostability in biological media is much more relevant for biomedical imaging.

Response: Thanks very much for the useful suggestion. We have revised Figure 11 (previous Figure 1f) in the revised manuscript as following:

Figure 11. Photostability comparison between ICG, IR26, BTC980, BTC982 and BTC1070 (5 μ M for all) in PBS (pH 7.4) under continuous-wave 808 nm exposure at a fluence rate of 2.3 W/cm².

11. Why does the emission spectrum of BTC1070 in Figure 5d not have a dip ~1150 nm while the emission spectrum in Figure 1d has a prominent dip ~1150 nm?

Response: The spectra difference at ~1150 nm of Figure 1j (previous Figure 1d) and Figure 5d is caused by the different absorption of dichloromethane and water. In fact, almost all fluorescent spectra have dips at different wavelengths due to the absorption of solvents. To make this point clearer for readers, we have tested the absorption spectra of several solvents used in this study. The results and discussion have been added to the revised Supporting Information as following:

Supplementary Figure 4. Absorption spectra of different solvents measured with a 2 mm (1 mm for H₂O) path length cuvette. CHCl₃: chloroform, DCM: dichloromethane, DCE: dichloroethane, MeCN: acetonitrile, DMSO: dimethyl sulfoxide, MeOH: methanol, H₂O: deionized water.

“As shown in Supplementary Figure 4, different solvents have complex fingerprint spectra in the entire NIR-II region. In our experiments, we used 10 mm path length cuvette for all the fluorescent spectra measurements. Thus, the influence of solvent absorption at 1000-1300 nm is unneglectable, which caused the different dips in fluorescent spectra (Supplementary Figure 2). As for water, there is no sharp peaks in the absorption band at 1000-1300 nm, which explains the smooth fluorescence spectra without prominent dips in Figure 5d and Supplementary Figure 3.”

12. A plot of the integrated NIR-II emission should be plotted as a function of pH. Based on Figure 3a's intensity scale and Figure 5d's emission spectrum, it seems that at physiological pH's the NIR-II intensity will be significantly lower than in acidic conditions. This probably explains why high 200 ms exposure times are needed for mouse hind-limb lymphatic imaging.

Response: We have added the following graph in the revised Supporting Information, showing the trends of the integrated emission intensity of BTC1070 as a function of pH.

Supplementary Figure 20. Plot of the integrated emission intensity of BTC1070 as a function of pH, corresponding to Figure 5d.

Indeed, the NIR-II intensity of BTC1070 at physiological pH is significantly lower than in acidic conditions. However, in PBS 7.4, BTC1070 is still ~5.7-fold brighter than ICG in the NIR-II (1000-1700) region. The main reason for the high 200 ms exposure times is that we used the emitted photons beyond 1200 nm under 1064 nm excitation for imaging, which account for only 8.1% of the total NIR-II photons. However, the reduced scattering and attenuation of long wavelength emission and excitation compensate for the low brightness and significantly improve image contrast in both phantom imaging and bioimaging.

13. The quantum yield in Table 1 needs to be listed in PBS in addition to DCM. Most other published NIR-II imaging agent quantum yields are listed in PBS.

Response: In the revised manuscript, we have listed the quantum yields of all BTC fluorophores in PBS in addition to DCM. The revised Table 1 is shown as follow.

Table 1 | Photophysical Properties of BTCs.

dye	λ_{abs}^a (nm)	ϵ ($\text{M}^{-1} \text{cm}^{-1}$)	λ_{fl}^a (nm)	Φ_{fl}^b (%)	$\epsilon\Phi_{\text{fl}}$ ($\text{M}^{-1} \text{cm}^{-1}$)
BTC980	932	184000	980	0.57/0.22 ^c	1049
BTC982	944	260000	982	0.68/0.3 ^c	1768
BTC1070	1014	115000	1070	0.09/0.016 ^c	104

^aPhotophysical properties in dichloromethane. ^bFor determination of the fluorescence quantum efficiency (Φ_{fl}), IR26 in dichloroethane ($\Phi_{\text{fl}} = 0.05\%$) was used as a fluorescence standard. ^cMeasured in PBS 7.4 in a phospholipid micelle formation.

14. From a quick literature search, the pH of mouse stomachs is ~4 which is still capable of being measured with BTC1070. This should be done as a control to show differences relative to simulated stomach acid.

Response: We have added the following graphs (Supplementary Figure 27 and S28) in the revised Supporting Information. As shown in Supplementary Figure 27, clear stomach profiles can be resolved with similar pseudo-color in ratio channel from $n = 3$ biologically independent normal mice. And the different pseudo-color compared with simulated groups in Figure 7 reveal a higher gastric pH in normal mice according to the color bar. Similar to the simulated groups, tissue coverage has negligible influence on the ratio values of normal group (Supplementary Figure 27d). Furthermore, we successfully resolved the pH values from ratiometric imaging by means of the calibration curves in Supplementary SI-7, which also show small ΔpH (< 0.27 unit) compared with standard pH meter results (Supplementary Figure 28, Supplementary Table 7). The results demonstrate BTC1070 is capable of noninvasively detecting gastric pH in normal mice ($n = 3$) by high-contrast ratiometric fluorescence imaging.

The following content has been added in the revised manuscript:

Page 8, line 4: “In addition, by the same imaging and quantification method, gastric pH in normal mice ($\text{pH} = 3.95 \pm 0.18$) can also be clearly visualized and measured with small ΔpH s lower than 0.27 pH unit (Supplementary Figure 27-28).”

Continued on Next Page

Supplementary Figure 27. **a,b,c**, Fluorescence images and corresponding ratiometric images of normal mice stomach gavaged with 20 μL BTC1070 micelle solution (dye concentration: 500 μM ; solvent: deionized water) at three imaging modes. **a**, Noninvasive imaging at $\sim 2\text{-}4$ mm tissue depth; **b**, Invasive imaging of gastric fluid covered by ~ 1 mm thickness of gastric wall; **c**, Imaging of exposed gastric fluid. The measurement of tissue depth/thickness is shown in Supplementary Figure 24. red channel: 900-1700 nm; cyan channel: 1000-1700 nm; ratio channel: $F_{1000\text{LP}}/F_{900\text{LP}}$. Color bar ranges from 5000 to 30000 for the red/cyan channel in a and b, 0 to 30000 for the red/cyan channel in c. The detailed imaging parameters for each image are listed in Supplementary Table 1. Imaging was performed independently on $n = 3$ biological replicates for each group with similar results. **d**, Ratiometric signals measured from the images. Data point with its error bar stands for mean \pm s.d. derived from $n = 3$ biologically independent mice.

Continued on Next Page

Supplementary Figure 28. Comparison of gastric pH values measured by ratiometric fluorescence imaging in Figure 27a, b, c and standard pH meter. Ratio values were converted to pH values by means of the calibration functions at corresponding depth (Supplementary SI-7). ΔpH = average pH resolved from ratiometric imaging - average pH measured by standard pH meter. Detailed pH data were summarised in Supplementary Table 7. Data point with its error bar stands for mean \pm s.d. derived from $n = 3$ biologically independent mice.

Supplementary Table 7. pH Values resolved from ratiometric images and measured by pH meter, corresponding to the results shown in Figure 7d and Supplementary Figure 23-25.

Analysis mode	pH Values		
	Low pH group	High pH group	Normal pH group
noninvasive	1.92 \pm 0.41	2.86 \pm 0.25	3.70 \pm 0.10
invasive	2.00 \pm 0.08	2.87 \pm 0.22	3.68 \pm 0.20
gastric fluid	2.16 \pm 0.07	3.10 \pm 0.16	3.80 \pm 0.12
pH meter	2.10 \pm 0.11	3.08 \pm 0.27	3.95 \pm 0.18 ^a

^aThe pH values were measured from the mice ($n = 3$) after gavage of fluorescent probes, which are independent to the normal pHs measured ($n = 6$) in the method section.

15. In the Discussion section, the authors should list possible steps to synthesize NIR-II pH sensitive probes that are sensitive near physiological pH values for ratiometric imaging.

Response: We have added the following contents in the Discussion section of the revised manuscript (Page 9, Line 17):

“In addition, affected by electron delocalization or electron-withdrawing effect of cyanine skeleton,

the pKa of BTC982 ($pK_{a1} = 0.72$) and BTC1070 ($pK_{a1} = 3.81$, $pK_{a2} = 0.29$) are both significantly lower than that of typical diaethylanilin analogs ($pK_a \approx 6.61$), far from physiological pH environment (6.5-7.4). However, this pKa lowering effect may inspire us to design physiological pH-responsive NIR-II fluorophores simply by replacing diethylamino with hydroxyl groups, because typical phenolic hydroxyl group has a higher pKa of ~ 10.0 .”

16. In the Discussion section, the authors should discuss possible steps to eliminate the need for micelle formulations. Micelle formulations hinder rapid excretion pharmacokinetics and are unlikely to yield renal clearance.

Response: We have added the following contents in the Discussion section of the revised Manuscript (Page 8, Line 30):

“Furthermore, another unresolved but important issue is the functionalization and hydrophilicity of these cyanine fluorophores. In this study, except for the classical functionalization route on polymethine chain, the flexible synthetic methods for benzothioopyryliums such as buchwald-hartwing amination (Scheme S2) also enable facile functionalization either on the N- or O-substituent moieties. And through further PEGylation or antibody/peptide bioconjugation, the possibly resulting improvements on hydrophilicity, pharmacokinetics, targeting or retention will expand more biological applications for these fluorophores.”

Reviewer #2 (Remarks to the Author):

This paper details the synthesis of NIR II fluorescent cyanines for biological imaging applications. The authors utilize a phenyl substituted benzothiopyrylium heterocycle to red-shift pentamethine cyanines into the second NIR window and compare the efficacy of these dyes versus the standard ICG using both the NIR I and NIR II windows for ICG.

The paper presents a wealth of experimental data for the new compounds; however there are major flaws which must be corrected before the manuscript can be accepted for publication.

Comments

- 1. First, the English language throughout needs thorough proof-reading. The manuscript is very difficult to follow at many points.***

Response: Thanks so much for the reviewer's comments. We have polished our manuscript carefully and corrected the grammatical, styling, and typos found in our manuscript. Detailed modifications are also provided as following:

Page 3, line 33 & Page 4, line 2: "showed" was changed to "show".

Page 4, line 3: "continues 808 nm laser" was changed to "continues-wave 808 nm laser".

Page 7, line 29: "with a tissue coverage depth of ~2-4 mm" was changed to "at a tissue depth of ~2-4 mm".

Page 8, line 18: "mehine" was changed to "methine"

Page 8, line 22: "other donor end-groups" was changed to "various cyanines with other heterocycles".

Page 8, line 29: "the charged π -system" was changed to "the charge of π -system".

Additionally, we have highlighted other modifications in the revised manuscript.

- 2. Throughout the manuscript, current NIR II dyes are described as "quenching absorbance" in polar solvents; quenching specifically refers to the reduction in fluorescence signal. The absorbance of these dyes broadens and undergoes a hypsochromic shift. Additionally, no mention of aggregation or other sources of this behavior are mentioned despite playing a significant role in the poor optical properties of cyanine dyes in aqueous solutions.***

Response: We agree with the reviewer's comment that quenching specifically refers to the reduction in fluorescence signal. However, the fluorescence of dye is mainly determined by two aspects, namely absorption coefficient at excitation wavelength and corresponding quantum yield. The reduction in any part will quench the fluorescence. In this study, we found that solvatochromism of

some commercial NIR-II cyanine fluorophores causes their significant attenuation in absorption. Especially for IR26, the weak absorption in aqueous solution leads to the significant fluorescence quenching as shown in following Figure R1.

Figure R1. Normalized fluorescence spectra of IR26 in DCM and H₂O (micelle formulation), showing the significant fluorescence quenching corresponding to the solvatochromism-caused quenching of absorption spectra in Figure 1c (previous Figure 1b).

We also strongly agree with the reviewer's comment that dye aggregation is another important factor for the poor optical properties of cyanine dyes in aqueous solution. Especially for cyanine with large π -conjugation, the hydrophobicity of main skeleton makes them easy to form H-aggregation (most of the time) in aqueous solution, which will lead to the broadening of absorption and decreasing of quantum yield. Therefore, in this study, we used micelle formulation to eliminate the influence from dye aggregation because the hydrophobic alkyl chains of phospholipids can act as steric barrier to prevent the dye aggregation, especially when the dye loading capacity is low.

We have added the following Supplementary Figure 3 in the revised Supporting Information. As shown in Supplementary Figure 3, with the decrease of dye loading capacity, the absorption as well as the fluorescence of BTCs increases significantly. The results indicate the influence of dye aggregation for fluorescence quenching can be eliminated when the dye loading capacity is lower than ~1wt%. However, IR26 shows weak absorption and small changes in fluorescence at the same condition, suggesting solvatochromism-caused quenching is the predominant factor instead of dye aggregation.

Figure R2. Absorption (a-d) and normalized fluorescence (e-h) spectra of IR26 (a, e), BTC980 (b, f), BTC982 (c, g) and BTC1070 (d, h) in PBS 7.4 (5 μ M for all). Dyes were loaded into phospholipid nanomicelles with varying loading capacity of 4%, 2% and 1%. Excitation wavelength: 808 nm. The results show that all dyes except for IR26 exhibit loading capacity dependent absorption and fluorescence spectra, which is a direct result of dye aggregation. IR26 shows weak absorption and small changes in fluorescence at the same condition, suggesting solvatochromism-caused quenching is the predominant factor instead of dye aggregation.

We have also added the following content in the revised manuscript:

Page 8, line 14: “Different from the commonly concept that aggregation causes fluorescence quenching in aqueous solution, such solvatochromism-caused quenching, unique to cyanine fluorophores, is significant even at a low dye-loading capacity of ~ 1 wt% in phospholipid micelle (Supplementary Figure 3).”

3. In the synthetic scheme, compounds 2 & 3c should have the diethylamino substituent, not the bromo group.

Response: We appreciate the reviewer for the useful suggestion. We have corrected this mistake in the revised Supporting Information.

4. In the absorbance spectra, all of the synthesized dyes have a small shoulder peak at ~ 1150 nm. This needs to be discussed in the text.

Response: Thanks so much for the useful suggestion. In the revised Supporting Information, we have a short discussion to this phenomenon. Overall, the shoulder peak at ~ 1150 nm or other wavelengths is caused by the absorption of different solvents. We tested the absorption spectra of several solvents used in this study (shown in Supplementary Figure 4), and added the following results and

discussion in the Supporting Information (Page S14).

Supplementary Figure 4. Absorption spectra of different solvents measured with a 2 mm (1 mm for H₂O) path length cuvette. CHCl₃: chloroform, DCM: dichloromethane, DCE: dichloroethane, MeCN: acetonitrile, DMSO: dimethyl sulfoxide, MeOH: methanol, H₂O: deionized water.

“As shown in Supplementary Figure 4, different solvents have complex fingerprint spectra in the entire NIR-II region. In our experiments, we used 10 mm path length cuvette for all the fluorescent spectra measurements. Thus, the influence of solvent absorption at 1000-1300 nm is unneglectable, which caused the different dips in fluorescent spectra (Supplementary Figure 2). As for water, there is no sharp peaks in the absorption band at 1000-1300 nm, which explains the smooth fluorescence spectra without prominent dips in Figure 5d and Supplementary Figure 3.”

5. For the computational modeling, BTC 982 shows a higher torsional angle between the two heterocycles; however, the discussion fails to postulate why this would be the case. Some discussion needs to be directed at explaining why this might be the case.

Response: We appreciate the reviewer for the useful suggestion. We have added the following discussion in the revised Supporting Information (Page S15, below the revised Supplementary Figure 5).

Supplementary Figure 5. Multi-view (a) images of optimized molecular geometries of BTC980 (b), BTC982 (c) and BTC1070 (d) in the ground state based on B3LYP/6-31G(d) level, showing the torsional angle between two benzothioapyrylium rings (shown in side view) as well as the distance between the two heterocyclic end-groups (shown in front view). Red dashed lines represent the face-to-face distance between two connected benzene rings, black solid lines represent the atom-to-atom distance between two connected hydrogen atoms.

“By measuring the distance between the two heterocycles for all three dyes, the shortest distance is ~6.72 Å in BTC982, located between the two terminal hydrogen atoms of diethylamino group and methoxy group, respectively. This result reveals that the two heterocycles of BTC982 have a larger intramolecular repulsion force, thus leading to a higher torsional angle between the two heterocycles.”

6. The NMR titrations for the pKa studies are confusing. The spectra for the two compounds do not appear to be substantially different; however, the authors conclude that one dye is protonating on the polymethine chain, then, upon reaching the second pKa that proton is transferred to the diethylamino group as the other amine protonates as well. This seems poorly supported by the titration studies and is rather difficult to see as being the case.

Response: The conclusion that BTC982 shows a methine-chain protonation feature is mainly derived from the acid-induced chemical shift changes of methine chain protons (H₆ and H₇ for BTC982, Supplementary Figure 21). **Upon the addition of deuterated TFA, the resonance peaks of methine chain protons (H₆ and H₇ in BTC982) show a process that disappears first and then reappears. By contrast, the similar methine chain protons of BTC1070 (H₅ and H₆ in BTC1070) are clearly observed from the spectra during the whole process (Supplementary Figure 22).**

These results provide a direct evidence for the methine-chain protonation feature of BTC982, because hydrogen-deuterium exchange induced by the methine-chain protonation will lead to the disappearance of proton resonances. This conclusion also can be identified by the chemical shift changes of methylene protons on diethylamino groups. The major downfield shifts of methylene protons of diethylamino groups on BTC982 occur in the TFA range of 1.5-25 μL , corresponding to the second protonation step on the nitrogen atoms, which is contrary to that of BTC1070.

We appreciate the reviewer for the useful comments. We have highlighted these main differences on the NMR titration spectra of BTC982 (Supplementary Figure 21) and BTC1070 (Supplementary Figure 22) in the revised Supporting Information. Furthermore, discussion about the NMR titration studies in the Supporting Information section (Page S29 & S30) have been revised as follows:

“A slight addition of deuterated TFA (0.5-1.5 μL) to BTC982 immediately led to the attenuation or even disappearance (at 1.5 μL) of resonance peaks corresponding to methine protons (H_6 and H_7), which was caused by the active hydrogen-deuterium exchange. Simultaneously, proton signals (H_{11}) of methylene near nitrogen atoms have little changes ($\Delta\delta[H_{11}] = 0.05$ ppm). The results indicate the protonation of BTC982 mainly occurred on the polymethine chain instead of the nitrogen atoms. When deuterated TFA was further added from 1.5 to 25 μL , the resonance peaks of methine protons gradually appeared ($\delta[H_6''] = 6.95$ ppm and $\delta[H_7''] = 7.21$ ppm) whilst proton resonances of methylene (H_{11}'') near nitrogen atoms shifted downfield obviously from 3.34 ppm to 3.61 ppm ($\Delta\delta[H_{11}'] = 0.27$ ppm). The results indicate that the addition of deuterium on polymethine chain was inhibited with the increase of acidity while the protonation site was transferred from methine units to nitrogen atoms, which led to the formation of a new cyanine structure (BTC982D_2^{2+}).”

“In contrast to BTC982, resonance peaks belong to the methine chain protons of BTC1070 can be clearly observed during the two protonation steps. Additionally, methylene protons of diethylamino groups show downfield shifts in the first protonation ($\Delta\delta[H_6] = 0.66$ ppm, $\Delta\delta[H_7] = 0.38$ ppm, $\Delta\delta[H_{11}] = 0.16$ ppm) as well as second protonation ($\Delta\delta[H_6'] = 0.46$ ppm, $\Delta\delta[H_7'] = 0.24$ ppm, $\Delta\delta[H_{11}'] = 0.13$ ppm), owing to the deshielding effect of protonated nitrogen atoms. The results imply the stepwise protonation process on nitrogen atoms.”

Reviewers' comments:

Reviewer #1 (Remarks to the Author):

The authors did a great job at addressing all reviewer comments. Minor grammatical errors still exist. Overall, the revised manuscript is an impressive scientific work and deserves publication in Nature Communications.

Reviewer #2 (Remarks to the Author):

The authors have responded to the previous round of critiques; however, I feel that some of their answers are incorrect and still need further work done.

The manuscript still has major issues with the English throughout. Additionally, the authors use of terms like quenching and solvatochromism are not quite correct. The argument that reducing the molar absorptivity also reduces the fluorescence does not make sense. The quantum yield of fluorescence remains the same; by their argument, reducing the concentration would be considered quenching as well.

Similarly, the authors use the term solvatochromism to describe the reduction in molar absorptivity cyanines experience in polar solvents. Solvatochromism specifically refers to a change in the λ_{max} due to changing solvent polarity. It does not describe hyper- or hypochromic shifts.

In Figure 1i, the experimental setup seems flawed. The light source is at the λ_{max} for ICG but the BTC dyes have peak absorbance higher than that. The authors should compare them using a broad-spectrum light source that would excite all dyes equally.

5. For the computational modeling, BTC 982 shows a higher torsional angle between the two heterocycles; however, the discussion fails to postulate why this would be the case. Some discussion needs to be directed at explaining why this might be the case.

Response: We appreciate the reviewer for the useful suggestion. We have added the following discussion in the revised Supporting Information (Page S15, below the revised Supplementary Figure 5).

Supplementary Figure 5. Multi-view (a) images of optimized molecular geometries of BTC980 (b), BTC982 (c) and BTC1070 (d) in the ground state based on B3LYP/6-31G(d) level, showing the torsional angle between two benzothioapyrylium rings (shown in side view) as well as the distance between the two heterocyclic end-groups (shown in front view). Red dashed lines represent the face-to-face distance between two connected benzene rings, black solid lines represent the atom-to-atom distance between two connected hydrogen atoms.

“By measuring the distance between the two heterocycles for all three dyes, the shortest distance is ~6.72 Å in BTC982, located between the two terminal hydrogen atoms of diethylamino group and methoxy group, respectively. This result reveals that the two heterocycles of BTC982 have a larger intramolecular repulsion force, thus leading to a higher torsional angle between the two heterocycles.”

I don't believe that that is a satisfactory answer. What program did the authors use to calculate these geometries and how were the compounds drawn prior to minimization/calculations? It seems odd that BTC982 is the only structure where the heterocycles are trans to each other while BTC980 and BTC1070 both have the heterocycles in a cis conformation. If the structure was drawn that way prior to minimization/calculations, then, depending on the program used, the calculations would not try to flip the heterocycle to the trans configuration. If that is indeed the correct ground state geometry for BTC982, then the authors should show the calculated relative energies of all three dyes in both the cis and trans heterocycle conformations to support BTC982 being the only one to adopt that conformation.

6. The NMR titrations for the pKa studies are confusing. The spectra for the two compounds do not appear to be substantially different; however, the authors conclude that one dye is protonating on the polymethine chain, then, upon reaching the second pKa that proton is transferred to the diethylamino group as the other amine protonates as well. This seems poorly supported by the titration studies and is rather difficult to see as being the case.

Response: The conclusion that BTC982 shows a methine-chain protonation feature is mainly derived from the acid-induced chemical shift changes of methine chain protons (H6 and H7 for BTC982, Supplementary Figure 21). **Upon the addition of deuterated TFA, the resonance peaks of methine chain protons (H6 and H7 in BTC982) show a process that disappears first and then reappears. By contrast, the similar methine chain protons of BTC1070 (H5 and H6 in BTC1070) are clearly observed from the spectra during the whole process (Supplementary Figure 22). These results provide a direct evidence for the methine-chain protonation feature of BTC982,** because hydrogen-deuterium exchange induced by the methine-chain protonation will lead to the disappearance of proton resonances. This conclusion also can be identified by the chemical shift changes of methylene protons on diethylamino groups. The major downfield shifts of methylene protons of diethylamino groups on BTC982 occur in the TFA range of 1.5-25 μL , corresponding to the second protonation step on the nitrogen atoms, which is contrary to that of BTC1070. We appreciate the reviewer for the useful comments. We have highlighted these main differences on the NMR titration spectra of BTC982 (Supplementary Figure 21) and BTC1070 (Supplementary Figure 22) in the revised Supporting Information. Furthermore, discussion about the NMR titration studies in the Supporting Information section (Page S29 & S30) have been revised as follows:

"A slight addition of deuterated TFA (0.5-1.5 μL) to BTC982 immediately led to the attenuation or even disappearance (at 1.5 μL) of resonance peaks corresponding to methine protons (H6 and H7), which was caused by the active hydrogen-deuterium exchange. Simultaneously, proton signals (H11) of methylene near nitrogen atoms have little changes ($\Delta\delta[\text{H11}] = 0.05 \text{ ppm}$). The results indicate the protonation of BTC982 mainly occurred on the polymethine chain instead of the nitrogen atoms. When deuterated TFA was further added from 1.5 to 25 μL , the resonance peaks of methine protons gradually appeared ($\delta[\text{H6}'] = 6.95 \text{ ppm}$ and $\delta[\text{H7}'] = 7.21 \text{ ppm}$) whilst proton resonances of methylene (H11') near nitrogen atoms shifted downfield obviously from 3.34 ppm to 3.61 ppm ($\Delta\delta[\text{H11}'] = 0.27 \text{ ppm}$). The results indicate that the addition of deuterium on polymethine chain was inhibited with the increase of acidity while the protonation site was transferred from methine units to nitrogen atoms, which led to the formation of a new cyanine structure (BTC982D₂²⁺)."

"In contrast to BTC982, resonance peaks belong to the methine chain protons of BTC1070 can be clearly observed during the two protonation steps. Additionally, methylene protons of diethylamino groups show downfield shifts in the first protonation ($\Delta\delta[\text{H}6] = 0.66$ ppm, $\Delta\delta[\text{H}7] = 0.38$ ppm, $\Delta\delta[\text{H}11] = 0.16$ ppm) as well as second protonation ($\Delta\delta[\text{H}6'] = 0.46$ ppm, $\Delta\delta[\text{H}7'] = 0.24$ ppm, $\Delta\delta[\text{H}11'] = 0.13$ ppm), owing to the deshielding effect of protonated nitrogen atoms. The results imply the stepwise protonation process on nitrogen atoms."

I don't feel that these arguments fully agree with the experimental results. Yes, the chemical shifts for H₆ and H₇ disappear; however, so does the chemical shift for H₁ and H₄ and H₈ are strongly attenuated. If protonation of the bridge were the only thing happening, this would not be the case. Based on the author's explanation, it would seem that the majority of the protons on this molecule are exchangeable. This seems highly unlikely though, as, if the protons were exchanging with deuterons, the signal would not return upon the addition of yet more deuterons. Any exchangeable protons would simply disappear and not return.

I suspect that the behavior of the NMR titration experiments is due to the placement of the diethylamino groups on BTC982 and BTC1070, the resonance effects of the two protonations would be observed differently. For BTC982, the nitrogen atoms are directly conjugated to the chromophore; thus, the protonation of the nitrogen atom would cause perturbations to chemical shifts throughout the polymethine chain. BTC1070, on the other hand, lacks conjugation between the chromophore and the diethylamino groups which should lessen any impacts protonation would have on the polymethine chain chemical shifts.

An easy proof for this would be to see the effects of protonation on the absorbance spectra of BTC982 and BTC1070. If the polymethine chain is being protonated as is proposed in the manuscript, there should be a drastic drop in absorbance due to the destruction of the conjugated system; however, if no such change is detected, then it would stand to reason that it is the nitrogen being protonated.

Point-to-Point Response to Reviewer's Comments

Reviewer #1 (Remarks to the Author):

The authors did a great job at addressing all reviewer comments. Minor grammatical errors still exist. Overall, the revised manuscript is an impressive scientific work and deserves publication in Nature Communications.

Response: We appreciate the reviewer very much for your constructive comments and suggestions on our manuscript. We have polished our manuscript carefully again. Detailed modifications are provided as following or highlighted in the revised manuscript:

Page 2, line 25: “were” was changed to “are”.

Page 2, line 26: “while” was changed to “and”.

Page 4, line 21, Page 7, line 17: “depth” was changed to “depths”.

Page 7, line 30: “coverage” was changed to “thickness”

Reviewer #2 (Remarks to the Author):

The authors have responded to the previous round of critiques; however, I feel that some of their answers are incorrect and still need further work done.

Response: We appreciate the reviewer very much for your constructive comments and suggestions on our manuscript.

1. The manuscript still has major issues with the English throughout.

Response: We have polished our manuscript carefully again. Detailed modifications are provided as following or highlighted in the revised manuscript:

Page 2, line 25: “were” was changed to “are”.

Page 2, line 26: “while” was changed to “and”.

Page 4, line 21, Page 7, line 17: “depth” was changed to “depths”.

Page 7, line 30: “coverage” was changed to “thickness”

2. Additionally, the authors use of terms like quenching and solvatochromism are not quite correct. The argument that reducing the molar absorptivity also reduces the fluorescence does not make sense. The quantum yield of fluorescence remains the same; by their argument, reduc-

ing the concentration would be considered quenching as well.

Response: We compared the fluorescence intensity or brightness of all dyes at the same concentration in this work.

a) In Figure 1k, the brightness comparison shows that IR26 is almost invisible and BTCs are bright. Also, in Table R1 (or to see Supplementary Table 3), the brightness of IR26 in aqueous solution is significantly lower than that of BTCs. Despite the decline in fluorescence after all dyes are transferred from DCM to water, IR26 suffers from more significant reduction. Obviously, in this case, the reduction in molar absorptivity should be more responsible for this change. We termed the phenomenon as *quenching*, because *quenching* generally refers to any process which decreases the fluorescence intensity of a given substance. *Quenching* is a relative concept, so is the word “*anti-quenching*” we used here. We used them for describing the differences between IR26 and **BTCs**, and on this basis, we emphasized the advantages of **BTCs** and their design strategies in order to be accepted by broader readers.

Table R1. Photophysical Properties of IR-26, BTC980, BTC982 and BTC1070 in DCM and PBS.

Dye	Solvent	ϵ (M^{-1} cm^{-1})	Φ η (%)	$\epsilon\Phi\eta$ (M^{-1} cm^{-1})
IR26	DCM	15000	0.	46
		0	03	
	PBS	3750	0. 01	0.375
	Decreased by (-fold)	40	3	120
BTC980	DCM	18400	0.	1049
		0	57	
	PBS	79600	0. 22	175
	Decreased by (-fold)	2.3	2. 6	6
BTC982	DCM	26000	0.	1768
		0	68	
	PBS	13750	0. 3	412
	Decreased by (-fold)	1.9	2. 3	4.3
BTC1070	DCM	11500	0.	104
		0	09	
	PBS	45000	0. 016	7.2
	Decreased by	2.6	5.	14.4

- b) As far as we know, it seems that the term *solvatochromism* is not limited to describe wavelength shift. In this reference (*Chem. Rev.* 1994, **94**, 2319-2358, **Page 2322**), the authors said “*It has long been known that UV/vis/near-IR absorption spectra of chemical compounds may be influenced by the surrounding medium and that solvents can bring about a change in the position, intensity, and shape of absorption bands. Hantzschlater termed this phenomenon solvatochromism.*”. And in a newer book (*Solvents and Solvent Effects in Organic Chemistry, 4th Edition* 2011. ISBN: 978-3-527-32473-6, **Page 360**), the authors said “*The term solvatochromism is used to describe the pronounced change in position (and sometimes intensity) of a UV/Vis absorption band that accompanies a change in the polarity of the medium.*”. The author of these two references, Prof. Reichardt, is a recognized world leader in this field.

In addition, the term *solvatochromism* have been widely used in lots of researches on cyanine and merocyanine fluorophores (*J. Am. Chem. Soc.* 1997, **119**, 3253-3258; *J. Mol. Struct. THEOCHEM* 2006, **766**, 49-60; *Angew. Chem. Int. Ed.* 2016, **55**, 2470-2473; *J. Am. Chem. Soc.* 2010, **132**, 4328-4335). In this work, cyanine fluorophores not only suffer from reduction in molar absorptivity in polar solvent, but also show spectra shift and broadening. We appreciate the reviewer’s comments, however, to our best knowledge, we cannot find any more appropriate term to describe this phenomenon.

3. *Similarly, the authors use the term solvatochromism to describe the reduction in molar absorptivity cyanines experience in polar solvents. Solvatochromism specifically refers to a change in the λ_{max} due to changing solvent polarity. It does not describe hyper- or hypochromic shifts.*

Response: Please see our response to question #2.

4. *In Figure 1i, the experimental setup seems flawed. The light source is at the λ_{max} for ICG but the BTC dyes have peak absorbance higher than that. The authors should compare them using a broad-spectrum light source that would excite all dyes equally.*

Response: Thanks very much for the useful suggestion. We have corrected Figure 1i (we think the reviewer referred to Figure 1i because Figure 1i is the absorption spectra of BTC1070) in the revised manuscript as following. In the photostability experiment we use 808 nm for ICG, 915 nm for BTC980, 940 nm for BTC982, and 1064 nm for BTC1070 and IR26, ensuring all dyes can be

equally excited.

Figure 11. Photostability comparison of all fluorophores (5 μ M) in PBS (pH 7.4) under continuous-wave laser exposure (ICG: 808 nm, BTC980: 915 nm, BTC982: 940 nm, BTC1070 and IR26: 1064 nm) at a fluence rate of 2.3 W/cm².

5. *I don't believe that that is a satisfactory answer. What program did the authors use to calculate these geometries and how were the compounds drawn prior to minimization/calculations? It seems odd that BTC982 is the only structure where the heterocycles are trans to each other while BTC980 and BTC1070 both have the heterocycles in a cis conformation. If the structure was drawn that way prior to minimization/calculations, then, depending on the program used, the calculations would not try to flip the heterocycle to the trans configuration. If that is indeed the correct ground state geometry for BTC982, then the authors should show the calculated relative energies of all three dyes in both the cis and trans heterocycle conformations to support BTC982 being the only one to adopt that conformation.*

1) **Response:** Thanks very much for the useful suggestion. We did all the quantum chemical calculations with Gaussian 09 suite (see Experimental section in Supplementary information, Page S5). All the 3D chemical structures of BTC fluorophores were drawn with ChemBio3D Ultra 13.0, and did MM2 energy minimization prior to Gaussian calculations. We have optimized the ground states of all BTC fluorophores either in the *cis*- or *trans*- conformation based on B3LYP/6-31G(d) level. As shown in the following results, *cis*-BTC980, *trans*-BTC982 and *cis*-BTC1070 have lower total energies compared with their isomers (Table R2). Similarly, the torsional angles between two heterocycles in all BTCs are smaller in their preferential conformations. The reason for the *trans*-conformation preferential in BTC982 may be related to the larger π -electron delocalization. We focused on the BTC1070 compound for the bioapplications because of its superior performance. Furthermore, this work involves broad disciplines including the lymphatic imaging and pH biosensing performance. We hope the reviewer could understand that it's difficult to just focus on dye chemistry here. We appreciate the reviewer for this comment, we realize that the mechanism behind still needs to be further studied in the future work.

Table R2. Calculated total energy and torsional angle of optimized molecular geometries of BTC fluoro-

phores.

Dye	Total energy (hartree)	Torsional angle (degree)
BTC980	cis -2373.99578905	15.48
	trans -2373.99375978	27.34
BTC982	cis -2797.77667278	41.00
	trans -2799.20233433	29.35
BTC1070	cis -2799.19409362	18.59
	trans -2799.19115027	33.72

Supplementary Figure 5. Multi-view (a) images of optimized molecular geometries of BTC980 (b), BTC982 (c) and BTC1070 (d) in the ground state based on B3LYP/6-31G(d) level, showing the torsional angle between two benzothiopyrylium rings (side view). The average lengths of C-N bonds connected to the conjugated system (black arrows) are measured to be 137.3 pm for BTC982 and 138.4 pm for BTC1070.

Figure R2. Multi-view images of optimized molecular geometries of *trans*-BTC980.

Figure R3. Multi-view images of optimized molecular geometries of *cis*-BTC982.

Figure R4. Multi-view images of optimized molecular geometries of *trans*-BTC1070.

6. *I don't feel that these arguments fully agree with the experimental results. Yes, the chemical shifts for H₆ and H₇ disappear; however, so does the chemical shift for H₁ and H₄ and H₈ are strongly attenuated. If protonation of the bridge were the only thing happening, this would not be the case. Based on the author's explanation, it would seem that the majority of the protons on this molecule are exchangeable.*

This seems highly unlikely though, as, if the protons were exchanging with deuterons, the signal would not return upon the addition of yet more deuterons. Any exchangeable protons would simply disappear and not return.

I suspect that the behavior of the NMR titration experiments is due to the placement of the diethylamino groups on BTC982 and BTC1070, the resonance effects of the two protonations would be observed differently. For BTC982, the nitrogen atoms are directly conjugated to the chromophore; thus, the protonation of the nitrogen atom would cause perturbations to chemical shifts throughout the polymethine chain. BTC1070, on the other hand, lacks conjugation between the chromophore and the diethylamino groups which should lessen any impacts protonation would have on the polymethine chain chemical shifts.

An easy proof for this would be to see the effects of protonation on the absorbance spectra of BTC982 and BTC1070. If the polymethine chain is being protonated as is proposed in the man-

uscript, there should be a drastic drop in absorbance due to the destruction of the conjugated system; however, if no such change is detected, then it would stand to reason that it is the nitrogen being protonated.

Response: This is a valid critique and we greatly appreciate the reviewer's helpful suggestion. We are very sorry for our incorrect conclusion about the protonation mechanism of BTC982. After reanalyzing the data, BTC982 may have the same protonation mechanism with BTC1070. The "disappeared" proton signals of methine chain (H₆ and H₇) may be caused by the perturbation of D-TFA acting as an ion-pairing reagent and the limitation of instrumental resolution.

We make the following revision to the manuscript and supplementary information:

Page 2, line 33 in Manuscript: The sentence "*Additionally, the protonation study shows the different protonation mechanisms for the diethylamino-substituted BTCs, revealing the different delocalization effects of nitrogen atoms at different position of terminal heterocycles.*" was revised as "*Additionally, the substituent position of diethylamino groups show significant influence on the photophysical properties and protonation behaviors of BTCs, revealing the different delocalization effects along the π -conjugation.*"

Page 9 in Manuscript: The discussion about the protonation studies of BTC982 and BTC1070 was reorganized (highlight region).

Supplementary Information: We have added the following results in the revised Supplementary Information SI-5.

Supplementary Figure 21. a, ¹H NMR spectra (400 MHz, CD₃CN) of BTC982 upon addition of different amount of deuterated TFA in 0.5 mL CD₃CN. The ratio between the integrated areas of signals H₁' , H₆' , H₇' and H₁₁' is roughly equal to 2:2:1:8. **b,** Proposed protonation mechanism for BTC982. Orange rectangles represent protonated regions.

Supplementary Figure 22. Absorption spectra of BTC982 (A) and BTC1070 (B) in MeCN (2 mL) upon the addition of different volumes of TFA.

As shown in Supplementary Figure 22, the absorption spectra changes of BTC982 upon addition of TFA reveal that there is only one protonation occurred in the TFA range of 0-100 μL (0 - 5% v/v in MeCN). This means that the NMR titration result of BTC982, in which D-TFA has a same concentration range (0-25 μL D-TFA in \sim 0.5 mL MeCN), also can only reveals the first protonation. Obviously, the acid of TFA/MeCN mixture cannot reach the second pK_a of BTC982, according to the absorption spectra in 6 M and 12 M HCl solution (Supplementary Figure 17).

Furthermore, Supplementary Figure 22a gives an isosbestic point at ~ 755 nm, indicating that the first protonation of BTC982 involves an equilibrium between two species. The NMR titration result of BTC982 (25 μ L D-TFA) clearly shows the methine chain protons' signals (H_1' , H_6' and H_7'), and the ratio between the integrated areas of these signals and the integrated area of a reference signal (H_{11}') is roughly equal to 2:2:1:8 (Supplementary Figure 21). Based on these results, the first protonation of BTC982 mainly occurred on one of the nitrogen atoms, same as BTC1070. Similarly, the corresponding absorption spectra behavior is also caused by the unsymmetrical cyanine structure.

The absorption spectra of BTC1070 upon the addition of TFA (Supplementary Figure 22b) also show two protonation processes. The two isosbestic points at ~ 850 nm and ~ 790 nm, respectively, reveal that both protonation steps involve equilibriums between two species. Combining with the NMR results, the protonation of BTC1070 is indeed occurred on the two nitrogen atoms step by step.

Overall, we have tried our best to study the protonation mechanism of BTC982. According to the data we collected currently, BTC982 and BTC1070 should have the same protonation mechanism, but exhibit different protonation behaviors. The pK_a values of BTC982 ($pK_{a1} = 0.72$, pK_{a2} not observed) are obviously lower than BTC1070 ($pK_{a1} = 3.81$, $pK_{a2} = 0.29$).

REVIEWERS' COMMENTS:

Reviewer #2 (Remarks to the Author):

The authors answered all the comments satisfactorily and the manuscript is acceptable to be published.

Thanks

Maged Henary, Ph.D.
Associate Professor
Department of Chemistry
Georgia state University
Atlanta, GA 30303